# The evolution of olfactory sensitivity, preferences, and behavioral responses in Mexican cavefish is influenced by fish personality

Maryline Blin, Louis Valay, Manon Kuratko, Marie Pavie, Sylvie Rétaux*

Paris-Saclay Institute of Neuroscience, CNRS and University Paris-Saclay, Saclay, France

*For correspondence:
sylvie.retaux@cnrs.fr

**Abstract** Animals are adapted to their natural habitats and lifestyles. Their brains perceive the external world via their sensory systems, compute information together with that of internal states and autonomous activity, and generate appropriate behavioral outputs. However, how do these processes evolve across evolution? Here, focusing on the sense of olfaction, we have studied the evolution in olfactory sensitivity, preferences, and behavioral responses to six different food-related amino acid odors in the two eco-morphs of the fish *Astyanax mexicanus*. To this end, we have developed a high-throughput behavioral setup and pipeline of quantitative and qualitative behavior analysis, and we have tested 489 six-week-old *Astyanax* larvae. The blind, dark-adapted morphs of the species showed markedly distinct basal swimming patterns and behavioral responses to odors, higher olfactory sensitivity, and a strong preference for alanine, as compared to their river-dwelling eyed conspecifics. In addition, we discovered that fish have an individual 'swimming personality', and that this personality influences their capability to respond efficiently to odors and find the source. Importantly, the personality traits that favored significant responses to odors were different in surface fish and cavefish. Moreover, the responses displayed by second-generation cave × surface F2 hybrids suggested that olfactory-driven behavior and olfactory sensitivity is a quantitative genetic trait. Our findings show that olfactory processing has rapidly evolved in cavefish at several levels: detection threshold, odor preference, and foraging behavior strategy. Cavefish is therefore an outstanding model to understand the genetic, molecular, and neurophysiological basis of sensory specialization in response to environmental change.

### eLife assessment

In this **important** paper, Blin and colleagues develop a high-throughput behavioral assay to test spontaneous swimming and olfactory preference in individual Mexican cavefish larvae. The authors present **compelling** evidence that the surface and cave morphs of the fish show different olfactory preferences and odor sensitivities and that individual fish show substantial variability in their spontaneous activity that is relevant for olfactory behaviour. The paper will be of interest to neurobiologists working on the evolution of behaviour, olfaction, and the individuality of behaviour.

## Introduction

With more than 26,000 species representing half of vertebrates, bony fishes are extremely diverse and have colonized all possible ecological niches (*Helfman et al., 2009*), making them outstanding models to study the neurophysiological, genetic, and evolutionary underpinnings of adaptive behaviors. To

colonize, survive, and thrive in various environments, sensory systems are particularly crucial as they serve as windows to the external world, and they display an exceptional diversity in fishes. From the visual system of deep-sea fish shaped to catch the rare photons or the point-like bioluminescent signals to the mobile barbels covered with taste buds to probe the sea bottom and detect buried prey of bottom feeders, examples abound (*de Busserolles et al., 2020*; *Kiyohara et al., 2002*).

The anatomical sensory specializations, i.e., the relative importance taken by specific sensory organs or brain sensory areas, arise during embryonic development through regulatory processes that control the patterning of the neuroepithelium and that define boundaries between presumptive neural regions (*Krubitzer et al., 2011*). The relative investment in different sensory modules governs anatomical and behavioral specializations. Among fish, sand-dwelling cichlids that rely almost exclusively on vision to execute behaviors develop a large optic tectum, whereas rock-dwelling cichlids, which inhabit complex environments, have small optic tecta but an enlarged telencephalon, and these variations arise early in development (*Sylvester et al., 2010*). Similarly, in the embryos of dark-adapted blind Mexican cavefish, the presumptive eyefield territory is reduced but the presumptive olfactory epithelium is increased in size, presumably as a compensation for the loss of visual modality in the dark environment (*Agnès et al., 2022*; *Hinaux et al., 2016a*; *Pottin et al., 2011*; *Torres-Paz et al., 2019*).

While the comparative anatomy of fish brains is reasonably well documented, behavioral studies have mainly focused on a few model species, primarily zebrafish (*Kalueff et al., 2013*). The behavioral correlates of sensory systems diversity and evolution in fishes are poorly described and understood, which hampers the interpretation of cross-species comparisons. For example, the size and complexity of fish olfactory organs is highly variable and correlates with the richness of odorant receptors repertoire, defining a morpho-genomic space in which olfactory specialists and non-specialists species distribute (*Burguera et al., 2023*; *Policarpo et al., 2022*). On the lower end, sunfish or pipefish have a small and flat olfactory epithelium and possess ~30 odorant receptor genes. On the higher end, polypteriforms have large and complex olfactory rosettes with up to 300 lamellae and possess ~1300 odorant receptor genes. Yet, the differences in olfactory behaviors, sensitivity, and preferences between these species are completely unknown.

To start addressing the question of the evolution of olfactory sensory-driven behaviors in an amenable laboratory fish model, we used the blind cave and the river-dwelling morphs of the Mexican tetra, the characiform *Astyanax mexicanus*. The species has become an established model for evolutionary biology at large, including evolution of behaviors (*Duboué and Keene, 2016*; *Hinaux et al., 2016b*; *Kowalko, 2020*; *Yoshizawa, 2016*). Cavefish embryos and larvae have larger olfactory pits than surface fish and their neuronal composition is changed (*Blin et al., 2018*; *Hinaux et al., 2016a*), while adults of the two forms have a similar olfactory rosette with 20–25 lamellae (*Schemmel, 1967*). The recent divergence between the two forms probably did not allow for a substantial evolution of their olfactory receptor gene repertoire, which is almost identical in the two eco-morphotypes (245 genes in cavefish, 233 genes in surface fish, from recent genome assemblies) (*Policarpo et al., 2022*). Yet, cavefish larvae tested in groups display outstanding olfactory detection capacities, as they are attracted to extremely low concentration of the amino acid alanine ($10^{-10}$ M) while surface fish larvae have a more 'classical' threshold ($10^{-5}$ M) (*Hinaux et al., 2016a*) which approximates levels of free amino acids found in natural waters (*Hara, 1994*). In wild natural caves too, groups of cavefish adults respond to low concentrations of odorants (*Blin et al., 2020*). We hypothesized that the evolution of olfactory system development and olfactory skills in blind cavefish is an adaptive trait that, together with other constructive sensory changes in mechano-sensory and gustatory systems (*Varatharasan et al., 2009*; *Yamamoto et al., 2009*; *Yoshizawa and Jeffery, 2011*; *Yoshizawa et al., 2014*), may have contributed to their adaptation and survival in the extreme environment of their subterranean habitat. Focusing on olfaction, here we developed a sensitive, high-throughput behavioral assay and a pipeline for analysis to compare the types of behavioral responses elicited by food-related, amino acid odors in cavefish and surface fish, at population and individual levels. We determined their olfactory preferences and sensitivity thresholds for six amino acids, and we analyzed their behavioral responses in detail. We also described olfactory-driven behaviors in second-generation (F2) hybrids resulting from crosses between surface and cave morphs, as an attempt to understand the genetic component of the 'olfaction trait' in *Astyanax* morphs. We discovered that the behaviors triggered by odorant stimulation has markedly evolved in cavefish.

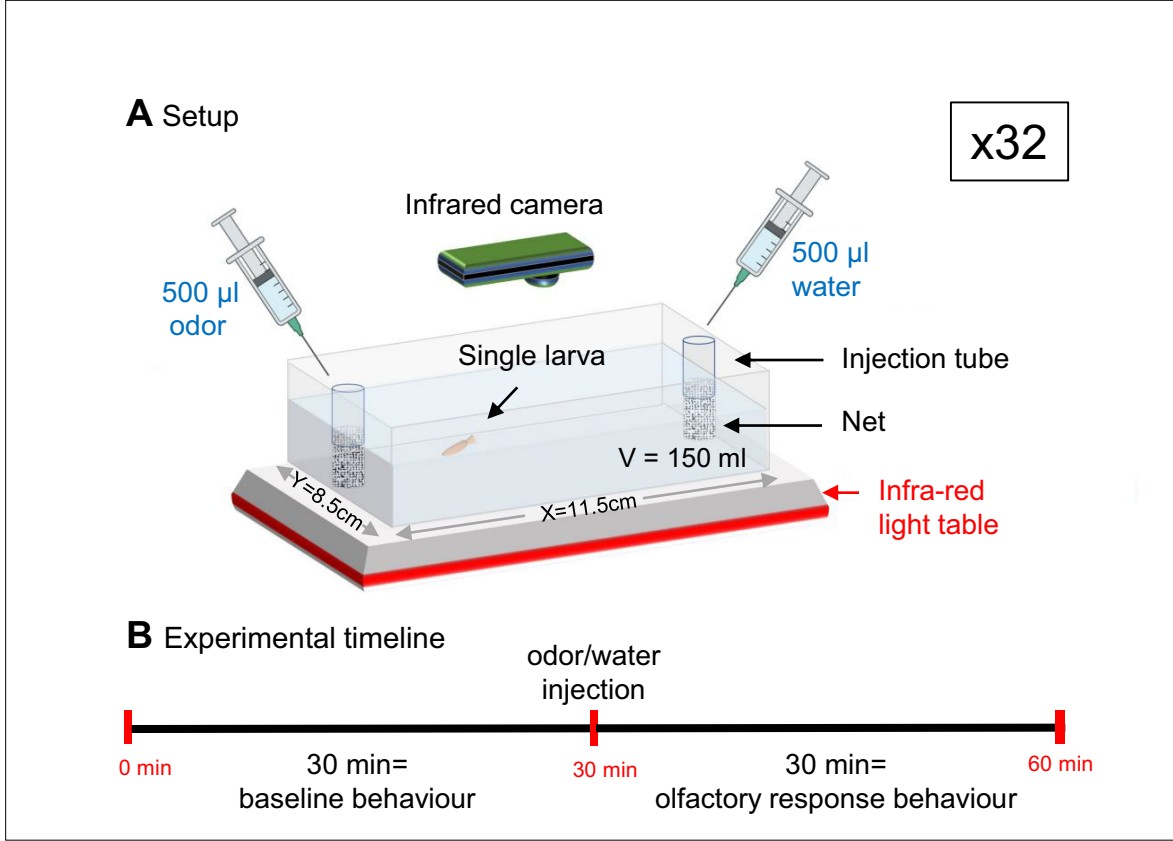

**Figure 1.** Experimental setup for testing behavioral responses to olfactory stimulation in individual, 6-week-old, *A. mexicanus* larvae. (**A**) Each fish is placed individually in 150 ml water in a rectangular test box placed on an infrared light emitting table. Control or odorant solution are delivered at the two extremities of the box, inside tubes covered with a net. Medium- to high-throughput behavioral testing is achieved by parallel recording of 32 test boxes placed under 8 infrared (IR) recording cameras (4 test boxes per camera). (**B**) After 1 hr or 24 hr habituation, the test consists of 1 hr IR video recording. The first 30 min provide the control/baseline behavior of individual fish. Odorant stimulation is given at 30 min, and the behavioral responses are recorded for 30 more minutes.

## Results

### A high-throughput, sensitive behavioral assay to compare individual's behaviors in *Astyanax* morphotypes

Our goal was to compare (1) olfactory discrimination capacities, (2) odor preferences, and (3) behavioral responses to olfactory stimuli in Pachón cavefish (CF), surface fish (SF), and F2 hybrids (F2) individuals. As the three types of fish are markedly different in terms of basal swimming activity and patterns, we first sought to characterize the diversity of these behaviors in order to be able to interpret accurately fish responses to odors.

Six-week-old fish were habituated either 1 hr or 24 hr in their individual test box and they were first recorded for 30 min in the dark, without any stimulus (*Figure 1*; see Methods). We observed and categorized several typical and distinctive 'baseline' swimming behaviors for individual fish (*Figure 2A*): random/haphazard swim (R), wall following (WF; defined as the fish continuously following the four sides of the box and turning around it, in a clockwise or counterclockwise fashion), large or small circles (C), thigmotactism (T, along the X- or the Y-axis of the box; defined as the fish swimming back and forth along one of the four sides of the box), or combinations thereof. The distribution of these different types of swimming patterns was significantly different in Pachón CF, SF, and F2 types of fish (*Figure 2B*; Fisher's exact test; see also Methods, *Figure 2—figure supplement 1* for another representation using correspondence analysis [CA] and *Figure 2—figure supplement 2* for a full color code representation of swimming patterns combinations). A majority of SF swam in a random pattern (blue shades), while a majority of CF performed wall following (red/brown shades) and F2 fish showed more diversified swimming patterns. Importantly, the distribution of these baseline-swimming patterns was

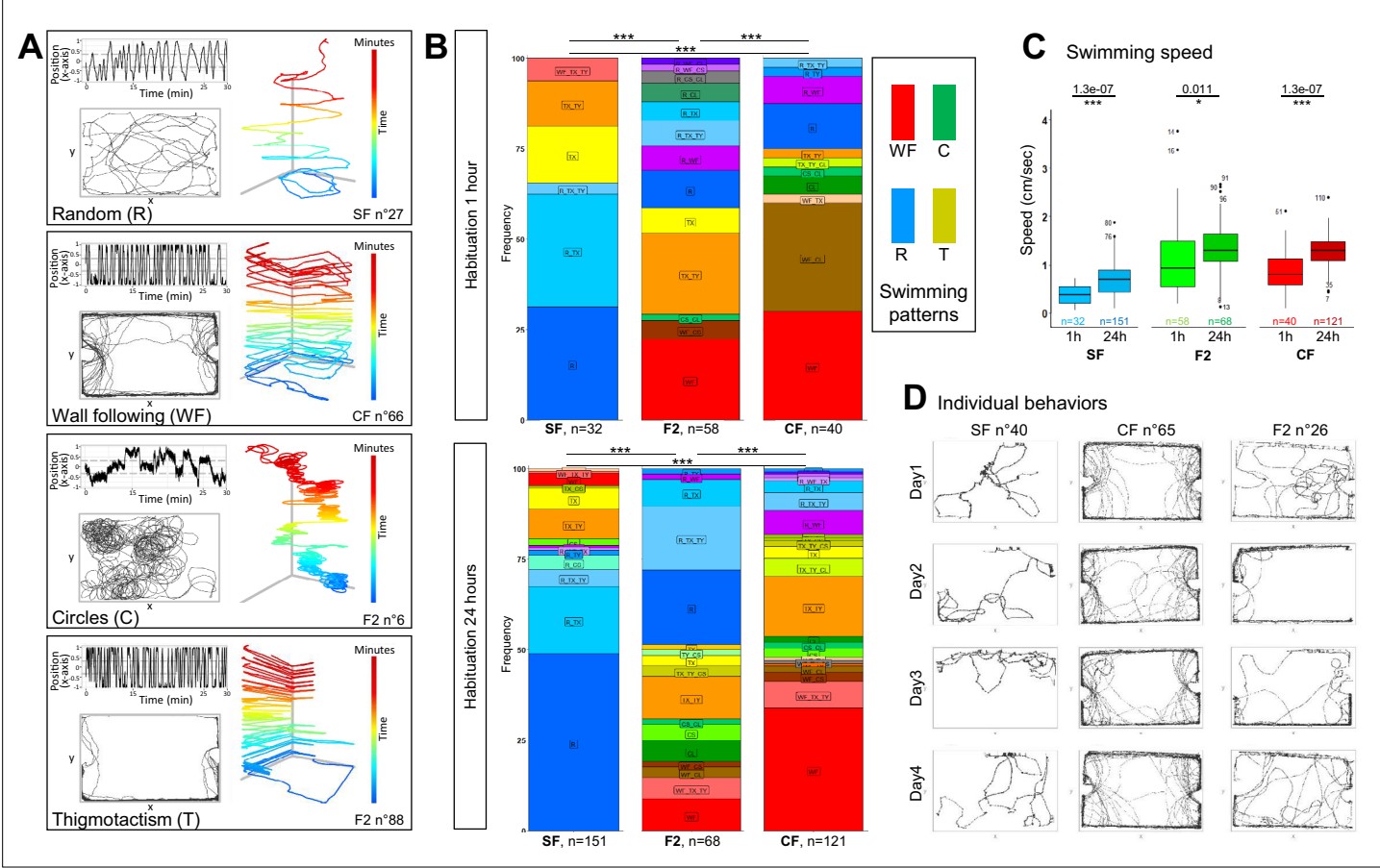

**Figure 2.** Basal swimming behaviors of 6-week-old cavefish (CF), surface fish (SF), and F2 hybrids (F2). (**A**) Examples of typical and distinctive basal swimming patterns exhibited by *A. mexicanus* morphotypes. The behavioral patterns (random, wall following, circles, thigmotactism) and the individual fish shown are indicated. Each pattern is best described by the combination of three types of representations. (1) The top left track shows the displacement of the fish along the x-axis of the box, representing back-and-forth swims along its long dimension. (2) The bottom left track is the top view, 2D representation of the trajectory. (3) The right track is the 2D plus time (color-coded) representation and helps, for example, to discern between wall following and thigmotactism. (**B**) Distribution of basal swimming pattern displayed by SF, F2, and CF after 1 hr or after 24 hr of habituation. Numbers of fish tested are indicated. The elementary color code is indicated (R in blue, WF in red, C in green, T in yellow). Exact swimming patterns and combinations are given on the colored plots (*Figure 2—figure supplement 2* for the full color code). Fisher's exact tests for statistical comparisons between groups. Of note, in this and subsequent figures, the swimming pattern color code does not relate whatsoever with the time color code used in the 2D plus time representation of swimming tracks such as in panel A. (**C**) Box plots showing the swimming speed in SF (blue), F2 (green), and CF (red), after 1 hr or 24 hr of habituation. Values are mean speed calculated over a period of 15 min. Numbers of fish tested are indicated. Mann-Whitney tests p-values are shown. (**D**) Examples of the stability of the basal swimming pattern over 4 experimental days in three individuals, with 24 hr of habituation time. One SF displaying random pattern, one CF displaying wall following, and one F2 displaying thigmotactism+random swim are shown.

The online version of this article includes the following source data and figure supplement(s) for figure 2:

**Source data 1.** Raw data file describing the mean swimming speed (averaged between the 10th and the 25th minute of recording) and the baseline swimming patterns of fish on the first day of experimental testing, after 1 hr or 24 hr of habituation.

**Source data 2.** Raw data file describing the baseline swimming patterns of fish on 4 consecutive days of experimental testing (d1 to d4), after 24 hr of habituation.

**Figure supplement 1.** Correspondence analyses (CA) for assessment of behavioral pattern differences between morphs and behavioral pattern change after different stimuli.

**Figure supplement 2.** Complete color code for the description of baseline swimming patterns.

**Figure supplement 3.** Stability of swimming patterns over time.

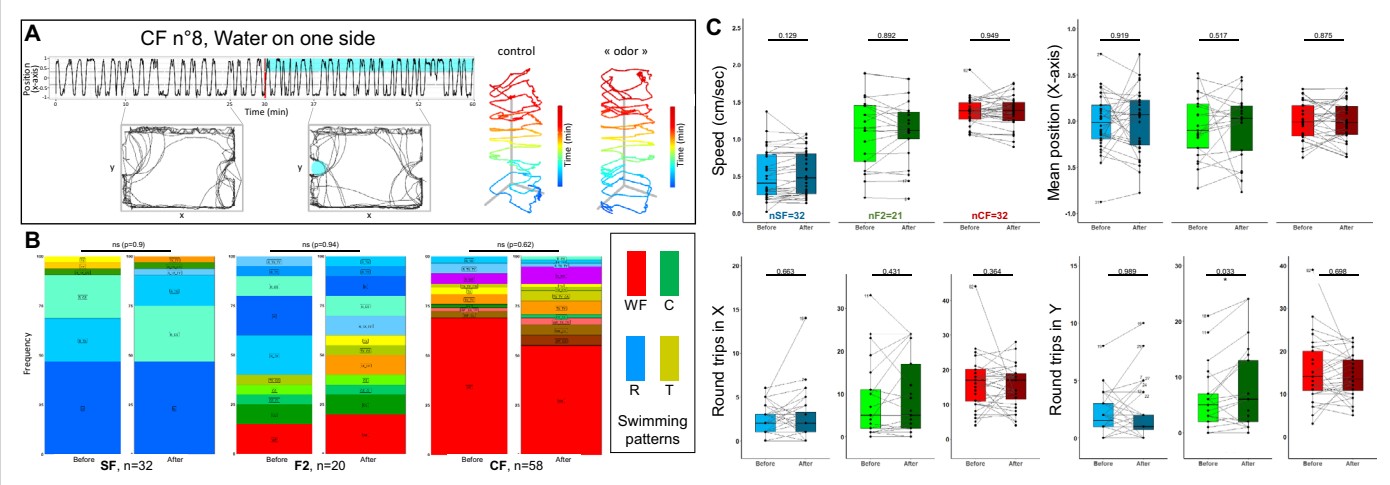

**Figure 3.** Controls on the olfactory setup and solution delivery method. (**A**) Lack of behavioral response in a representative cavefish (CF) individual after injection of water on one side of the test box. (**B**) Lack of change in qualitative swimming pattern displayed by surface fish (SF), CF, and F2 hybrids (F2), before and after water injection on one side. Fisher's exact tests. (**C**) Lack of change in quantitative swimming kinetics in SF (blue), CF (red), and F2 (green), before (pale colors) and after (dark colors) water injection on one side. In this and the following graphs, box plots show the swimming speed, the fish position along the X-axis of the test box, and the numbers of back-and-forth swims along the X (length) and Y (width) axes of the box. All values are averaged over a 15 min period, either before or after the injection. A thin line links the value 'before' and the value 'after', for each individual. Numbers of fish tested are indicated on the 'speed' box plots. Numbers next to dots indicate the identity of outlier individuals.

The online version of this article includes the following figure supplement(s) for figure 3:

**Figure supplement 1.** Additional controls for experimental design.

affected as a function of the habituation time, in all three types of fish (*Figure 2B*; p=0.044 for SF, p=0.0005 for CF, and p=0.0005 for F2, Fisher's exact tests). Moreover, thigmotactism behavior, which we have previously shown to represent stress behavior (*Pierre et al., 2020*), was frequent in SF after 1 hr habituation (70% of individuals) but reduced after 24 hr acclimation (37%). In the same line, the swimming speed was different for all fish types when computed after 1 hr or after 24 hr of habituation (*Figure 2C*). This suggested that natural, unperturbed behavior is better observed after long, 24 hr habituation, which we applied thereafter.

As described above at population level, CF, SF, and F2 displayed overall markedly different basal swimming patterns (*Figure 2B*; see also Methods and *Figure 2—figure supplement 1* for another representation using CA). Yet, at individual level, there was a substantial degree of variability across individuals within a morphotype. That is, even though most SF tended to swim randomly (75% after 24 hr habituation; blue shades on graphs) and most CF performed wall following (50%; red/brown shades), these behaviors were not exclusive as some SF individuals displayed thigmotactism behavior or some CF individuals swam in circles. Moreover and importantly, the type of swim pattern of each individual fish was remarkably stable and reproducible along several days of recordings, and swimming kinetics (position in the box, swimming speed, number of round trips in X and Y) showed no or very little variation (*Figure 2D* and *Figure 2—figure supplement 3*). Together, these observations strengthened the importance of (1) using long habituation times and (2) studying individual behaviors. They further highlighted an overlooked aspect of fish behavioral analyses: fish may have a 'personality' that we sought to take into account when comparing behavioral responses to odors below.

Finally, Pachón CF, SF, and F2 possess markedly different sensory apparatuses and capacities, not only for chemo- but also for mechano-sensation (*Lunsford et al., 2022*; *Yoshizawa et al., 2010*). Therefore, to test behavioral responses specific to olfactory stimuli, we wanted to find a way to deliver odors that would be vibration-less. We choose to inject odorant solutions inside tubes attached at the two extremities of the test box, designed to stop the wave and flow that would otherwise arise at the surface of the water upon injection (*Figure 1*). The efficacy of the device was demonstrated by the lack of perturbation in swimming behavior, neither qualitative (i.e. behavioral response and swimming pattern; *Figure 3AB*) nor quantitative (i.e. swimming kinetics; *Figure 3C*) for the three types of fish when water/control injections were performed on one side of the test box (or two sides;

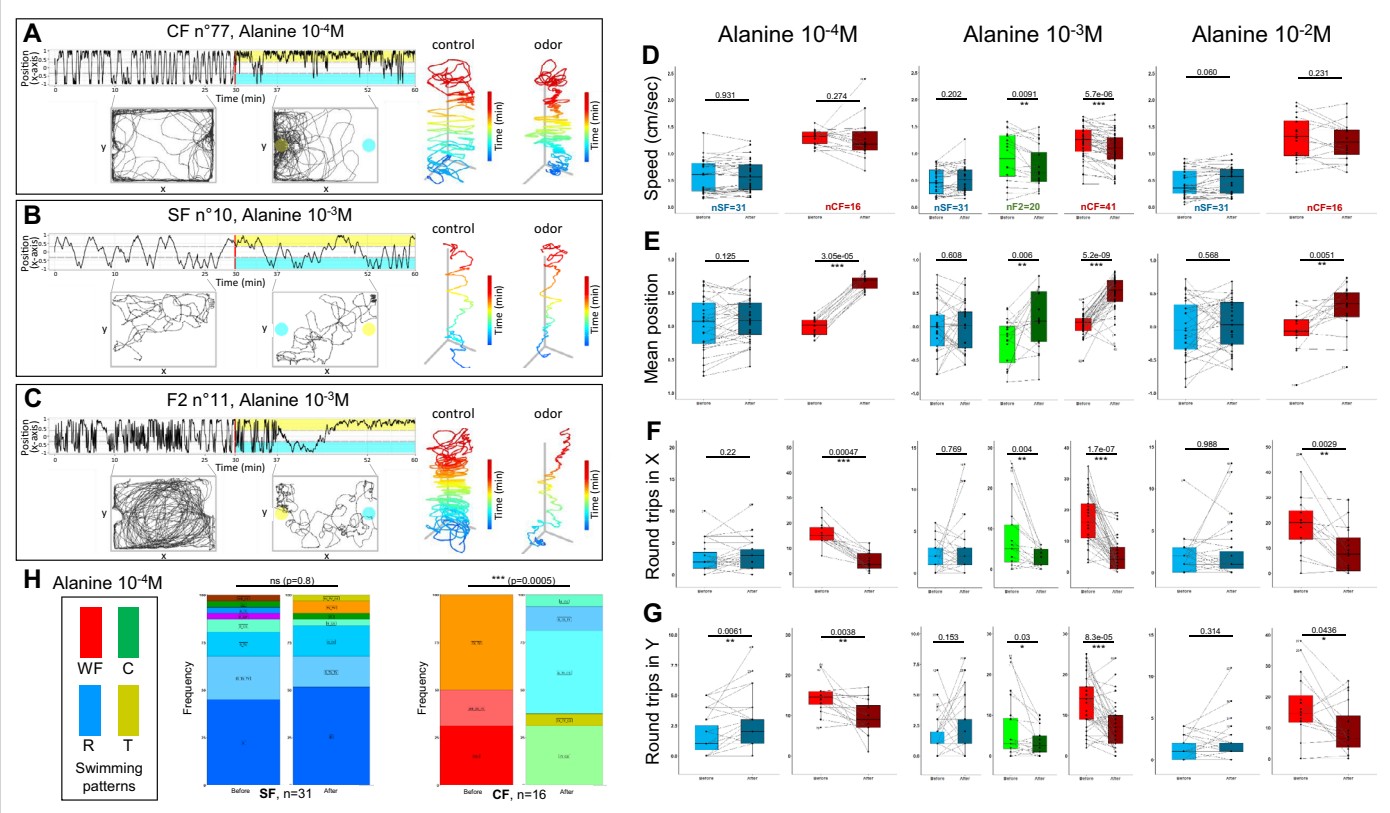

**Figure 4.** Behavioral responses to alanine of 6-week-old cavefish (CF), surface fish (SF), and F2 hybrids (F2). (**A–C**) Representative individual responses of CF, SF, and F2 after injection of alanine at the indicated concentrations. In the two left graphs of each panel, the blue color indicates the water/control injection side and the yellow color indicates the alanine injection side. (**D–G**) Box plots showing swimming speed (**D**), mean position along the X-axis (**E**) and the number of back-and-forth trips in X and Y (**F,G**) in SF (blue), CF (red), and F2 (green), before (lighter color) or after (darker color) the injection of alanine at the indicated concentration. Values are calculated over a 15 min period. Black lines link the 'before odor' and 'after odor' value for each individual. Numbers close to black dots indicate the identity of outlier individuals. p-Values from paired Mann-Whitney tests are shown. The number of fish tested is indicated. (**H**) Change in swimming patterns exhibited by SF and CF after injection of alanine $10^{-4}$ M. Fisher's exact tests.

The online version of this article includes the following source data and figure supplement(s) for figure 4:

**Source data 1.** Raw data file describing behavioral responses of fish to various concentrations of different amino acids.

**Figure supplement 1.** Additional examples of behavioral responses to alanine.

**Figure supplement 2.** Correspondence analyses (CA) for assessment of behavioral pattern change after alanine injection.

*Figure 3—figure supplement 1A and B*). This suggested that our way to deliver odorant solutions did not generate vibratory perturbations. Moreover, when no injection was made, individual fish behaviors were stable over 1 hr of test (*Figure 3—figure supplement 1C*), allowing to observe accurately the effects of an olfactory stimulus if applied after 30 min. Finally, all recordings were performed in the dark under infrared lights to neutralize the visual modality in sighted fish, a procedure that did not affect SF behavior (*Figure 3—figure supplement 1D*). Hence, the behavioral responses to odors recorded thereafter in Pachón CF, SF, and F2 are driven by olfaction and can be compared (see also *Hinaux et al., 2016a*).

## Behavioral responses to alanine in *Astyanax* morphotypes

Amino acids are food-related odorant cues for fish. Next, we characterized individual behavioral responses elicited by different concentrations of alanine, a potent aliphatic amino acid cue for most fish species including *Astyanax*.

Injection of alanine $10^{-2}$ M, $10^{-3}$ M, $10^{-4}$ M or $10^{-5}$ M (hence $10^{-4}$ M, $10^{-5}$ M, $10^{-6}$ M, or $10^{-7}$ M final concentration in the odorant third of the testing box, respectively) induced a strong behavioral response in cavefish (*Figure 4A*, $10^{-4}$ M injection shown; *Figure 4—figure supplement 1A* for other

examples). Upon stimulus, CF decreased the numbers of back-and-forth swims along the X- and the Y-axes of the box (*Figure 4A*, top left graph and *Figure 4FG*), and they shifted and restricted their swimming activity toward the odorant side of the arena, close to the odor delivery tube (*Figure 4A*, bottom left graph, and *Figure 4E*). A tendency to decreasing swimming speed was observed but was significant only for $10^{-3}$ M injections (*Figure 4D*). Alanine injection also markedly changed CF swimming patterns, as they completely switched from WF and T to R and C swimming modes (*Figure 4H* and *Figure 4—figure supplement 1D*; Fisher's exact tests p-values: 0.014, 0.0005, 0.0005, and 0.099 for $10^{-2}$ M, $10^{-3}$ M, $10^{-4}$ M, or $10^{-5}$ M alanine injections, respectively; see also Methods and *Figure 4—figure supplement 2* for another representation using CA). In sum, CF individuals displayed robust behavioral responses and attraction to alanine. Of note, further decreasing the concentration of injected alanine to $10^{-6}$ M, $10^{-7}$ M, $10^{-8}$ M, $10^{-9}$ M, and $10^{-10}$ M had milder to no effects on cavefish. Swimming patterns remained globally unchanged (not shown), but swimming speed was decreased (p=0.005 for $10^{-7}$ M; n=16), numbers of round trips in Y were decreased (p=0.012 for $10^{-7}$ M; n=16), and numbers of round trips in X were decreased (p=0.017 for $10^{-7}$ M; p=0.029 for $10^{-8}$ M; p=0.003 for $10^{-9}$ M; n=16 each; data not shown). This suggested that in the present experimental setup, cavefish can detect very low concentrations of alanine, in agreement with previously published data (*Hinaux et al., 2016a*).

Conversely, surface fish responses were more subtle and seemed restricted to some individuals (*Figure 4B*; *Figure 4—figure supplement 1B* for other examples). Notably, the numbers of back-and-forth swims along the X- and the Y-axes of the box were mostly unchanged and the position in the box did not vary for SF upon alanine injection (*Figure 4B*, top and bottom left graphs and *Figure 4E–G*). The swimming speed was unchanged (*Figure 4D*). Swimming patterns were globally unaffected upon alanine injection (*Figure 4H* and *Figure 4—figure supplement 1D*; Fisher's exact tests p-values: 0.16, 0.54, and 0.80 for $10^{-2}$ M, $10^{-3}$ M, or $10^{-4}$ M alanine injections, respectively; *Figure 4—figure supplement 2* for another representation using CA). As shown in *Figure 4—figure supplement 1B*, some rare SF individuals appeared to change their swimming mode upon alanine $10^{-2}$ M (high concentration) injection, suggesting that they did perceive and react to the odorant stimulus. However, these rare individual responses were 'diluted' at population level by the pooling of all fish in the distribution graphs. In summary, SF behavioral responses were modest and markedly different from CF.

Finally, F2 were tested with alanine $10^{-3}$ M, a concentration that elicits strong responses in CF but not in SF. Upon stimulus, F2's change in behaviors were similar to CF (*Figure 4C*; *Figure 4—figure supplement 1C* for other examples): they decreased back-and-forth swimming activity, decreased swimming speed, and swam close to the odor source (*Figure 4D–G*). They also shifted their swimming patterns (*Figure 4—figure supplement 1D*; Fisher's exact test p=0.0065), including a loss of wall following mode that was reminiscent of the trend observed in CF. As an illustration, the individual shown in *Figure 4C* displays a striking change from a 'large circles' to a 'random' swimming pattern accompanied with variations in swimming kinematics.

Overall, these data suggested that, compared to SF, CF detection threshold and behavioral responses to the amino acid alanine have significantly evolved.

## Behavioral responses to serine and cysteine in *Astyanax* morphotypes

We next systematically tested behavioral responses to injections of $10^{-2}$ M, $10^{-3}$ M, or $10^{-4}$M of serine (polar amino acid, hydroxyl group) and cysteine (non-polar, sulfur containing), two other potent amino acid olfactory cues for fish (*Figure 5* and *Figure 5—figure supplement 1*).

Serine elicited behavioral responses similar to alanine: CF (as well as F2 hybrids, *Figure 5—figure supplement 1A*) moved toward the odor source, whereas SF did not (*Figure 5A, B, and D*). Swimming speed was unchanged in all fish types (*Figure 5C*), but a decrease or an increase of back-and-forth swimming activity was observed in CF and SF, respectively (*Figure 5E and F*). At population level, significant changes in swimming patterns were observed for the three fish types upon $10^{-2}$ M serine (*Figure 5G*; Fisher's exact p-values 0.0005, 0.001, and 0.0085 for SF, CF, and F2, respectively; *Figure 5—figure supplement 2* for CA). Of note, and contrarily to alanine, the lower concentrations of serine tested ($10^{-4}$ M and $10^{-3}$ M in syringe; $10^{-6}$ M and $10^{-5}$ M in box, respectively) were unable to trigger a robust behavioral change in cavefish (not shown, see *Supplementary file 1A*), suggesting that their detection threshold for serine is higher (not as good as) than for alanine. Regarding SF, responses were also observed only for $10^{-2}$ M serine injections.

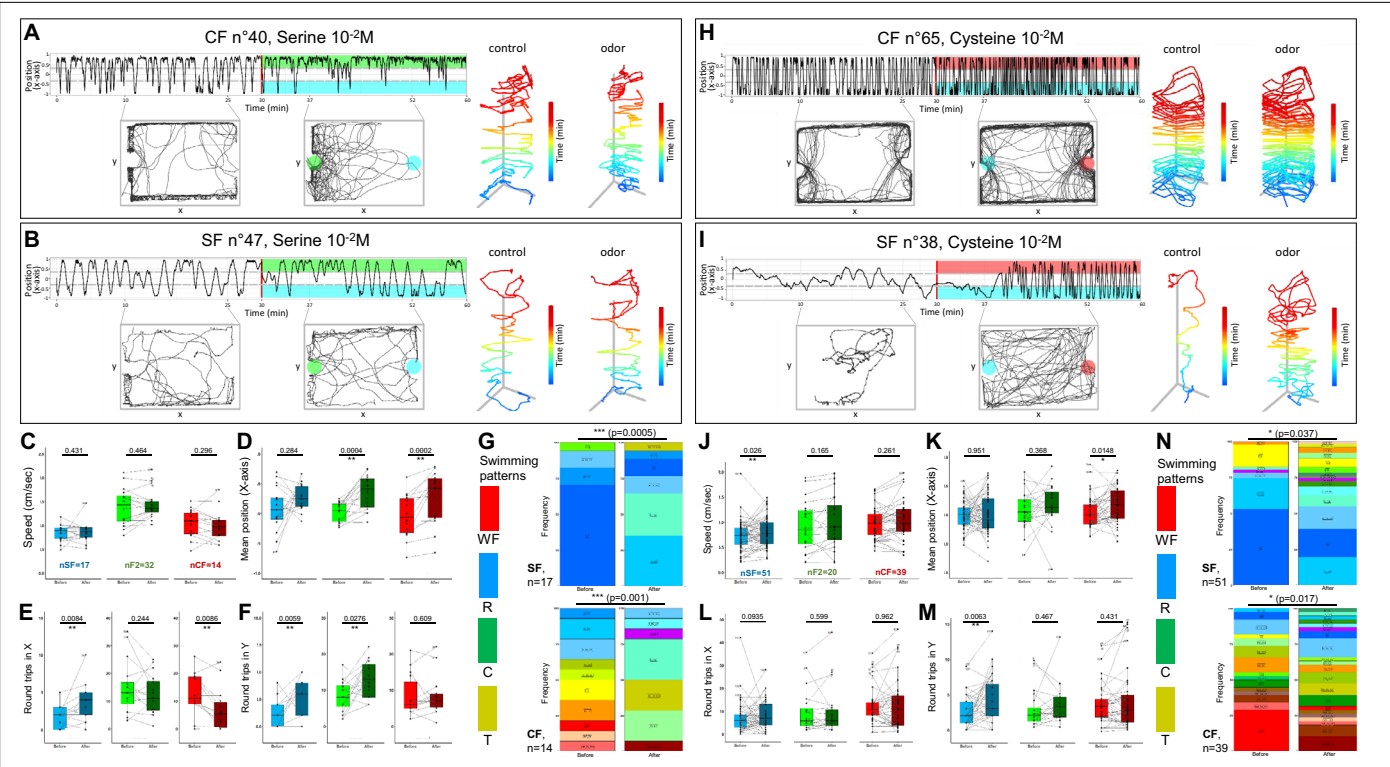

**Figure 5.** Behavioral responses to serine and cysteine 10⁻² M. (**A,B** and **H,I**) Representative individual responses of cavefish (CF) and surface fish (SF) after injection of serine (**A,B**) or cysteine (**H,I**) at high concentration (10⁻² M). In the two left graphs, the blue color indicates the water/control injection side and the green (serine) or red (cysteine) color indicates the injection side. (**C–F** and **J–M**) Box plots showing swimming speed (**C,J**), mean position along the X-axis (**D,K**), and the number of back-and-forth trips in X and Y (EL and FM) in SF (blue), CF (red), and F2 hybrids (F2) (green), before (lighter color) or after (darker color) injection of serine 10⁻² M (**C–F**) or cysteine 10⁻² M (**J–M**). Values are calculated over a 15 min period. Black lines link the 'before odor' and 'after odor' value of each individual fish. Numbers close to black dots indicate the identity of outlier individuals. p-Values from paired Mann-Whitney tests are shown. The number of fish tested is indicated. (**G** and **N**) Change in swimming patterns elicited after injection of 10⁻² M serine (**G**) or cysteine (**N**) in SF and CF (F2 not shown). Fisher's exact tests.

The online version of this article includes the following source data and figure supplement(s) for figure 5:

**Source data 1.** Raw data file describing behavioral responses of fish to various concentrations of different amino acids.

**Figure supplement 1.** Additional examples of behavioral responses to serine and cysteine.

**Figure supplement 2.** Correspondence analyses (CA) for assessment of behavioral pattern change after serine and cysteine injection.

Cysteine also produced significant effects (*Figure 5H–N* and *Figure 5—figure supplement 1B*). Swimming speed and numbers of back-and-forth swims along the Y-axis increased only in SF and for the three concentrations tested (*Figure 5J and M* for 10⁻² M) (for 10⁻³ M: p=0.01 and p=2e-05, n=32; for 10⁻⁴ M: p=2e-05 and p=0.00028, n=32). CF were attracted to the odorant side for the three concentrations tested (p=0.013–0.025; n=16–38 each), whereas SF did not change position (*Figure 5K*) except for cysteine 10⁻³ M to which they seemed to be repelled (p=0.0003, n=32). The three fish types changed significantly their swimming patterns (*Figure 5N*; Fisher's exact p-values 0.03, 0.017, and 0.04 for SF, CF, and F2, respectively; *Figure 5—figure supplement 2* for CA). After injection of cysteine at the lower concentrations of 10⁻³ M or 10⁻⁴ M, CF behavioral responses were similar to those observed for alanine and serine, i.e., a decrease in back-and-forth swimming activity along the X-axis and a significant change in position toward the odor source, without changing swimming speed (not shown, see *Supplementary file 1A*).

In sum, the different fish types show diverse responses to different concentrations of different amino acids.

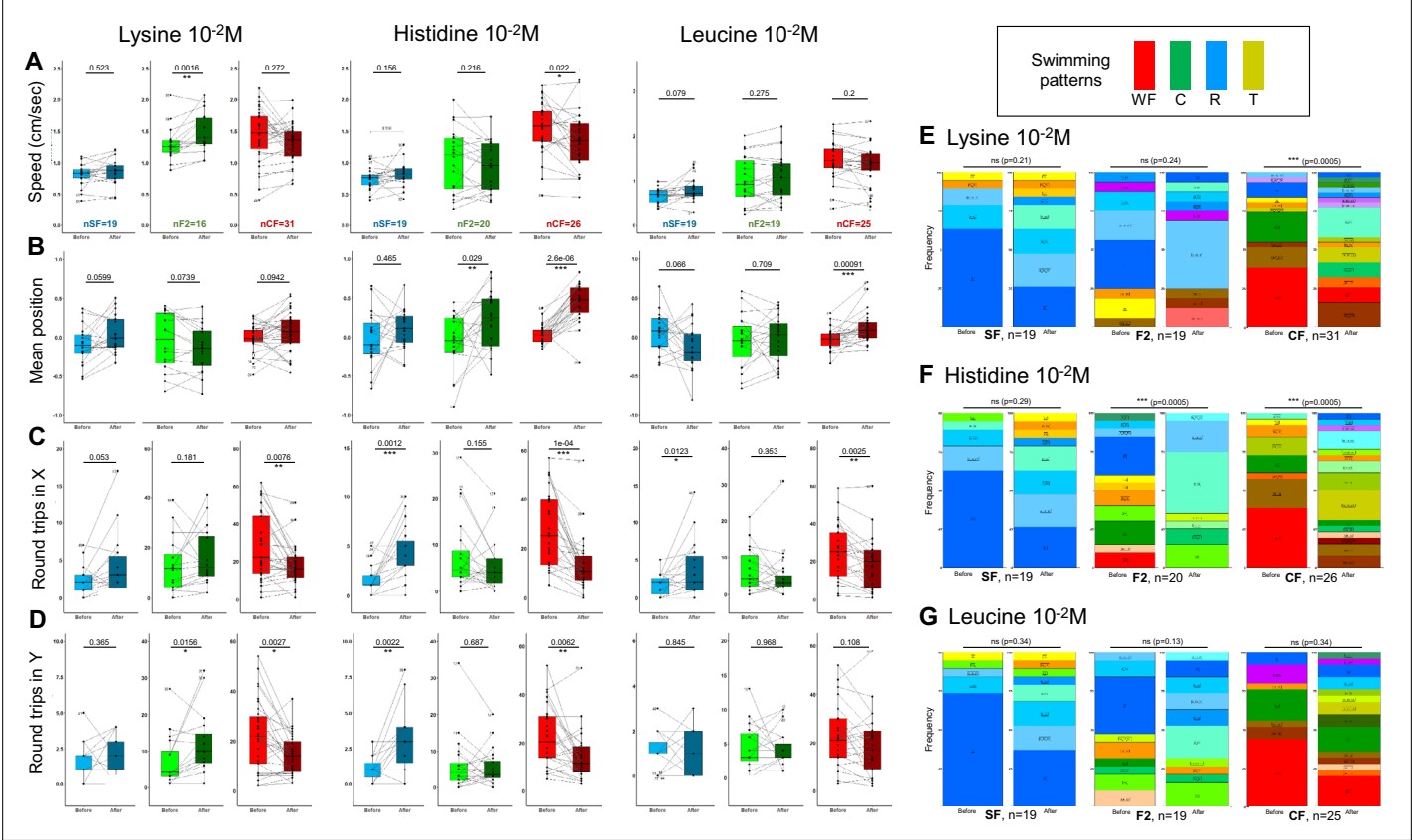

**Figure 6.** Behavioral responses to lysine, histine, and leucine $10^{-2}$ M. (**A–D**) Box plots showing swimming speed (**A**), mean position along the X-axis (**B**), and the number of back-and-forth trips in X and Y (CD) in surface fish (SF) (blue), cavefish (CF) (red), and F2 hybrids (F2) (green), before (lighter color) or after (darker color) the injection of the indicated amino acid. Values are calculated over a 15 min period. Black lines link the 'before odor' and 'after odor' value of each individual fish. Numbers close to black dots indicate the identity of outlier individuals. p-Values from paired Mann-Whitney tests are shown. The number of fish tested is indicated. See *Figure 6—figure supplement 1* for examples of representative individual responses. (**E–G**) Change in swimming patterns elicited after injection of $10^{-2}$ M lysine (**E**) or histidine (**F**) or leucine (**G**) in SF, CF, and F2. Fisher's exact tests.

The online version of this article includes the following source data and figure supplement(s) for figure 6:

**Source data 1.** Raw data file describing behavioral responses of fish to various concentrations of different amino acids.

**Figure supplement 1.** Examples of behavioral responses to lysine (ABC), histidine (DEF), and leucine (GHI) $10^{-2}$ M (high concentration).

**Figure supplement 2.** Correspondence analyses (CA) for assessment of behavioral pattern change after lysine, histidine, and leucine injection.

## Behavioral responses to lysine, histidine, and leucine in *Astyanax* morphotypes

Finally, we choose to examine responses triggered by high concentrations ($10^{-2}$ M) of three other, less studied amino acids: lysine, histidine (both polar and positively charged), and leucine (aliphatic, like alanine) (*Figure 6* and *Figure 6—figure supplement 1*).

For these three odors, CF responses were conspicuous and could include a shift of swim position toward the odor's source, a decrease of back-and-forth swimming activity with decrease in swimming speed, and significant changes in swim patterns at population level (except for leucine) (*Figure 6A–G* and *Figure 6—figure supplement 1A, D, G*; *Figure 6—figure supplement 2* for CA). Regarding SF, changes were restricted to increases in back-and-forth swimming activity for histidine, without change in swimming speed or position in the box (*Figure 6A–G* and *Figure 6—figure supplement 1B, E, H*). These three amino acids did not elicit changes in swimming pattern in SF at population level (*Figure 6E–G*). Finally, F2 also showed specific qualitative (swimming patterns) and quantitative responses to each of these three amino acids (*Figure 6A–G* and *Figure 6—figure supplement 1C, F, I*). Together, these data suggested that CF, SF, and F2 detected and responded in their own and

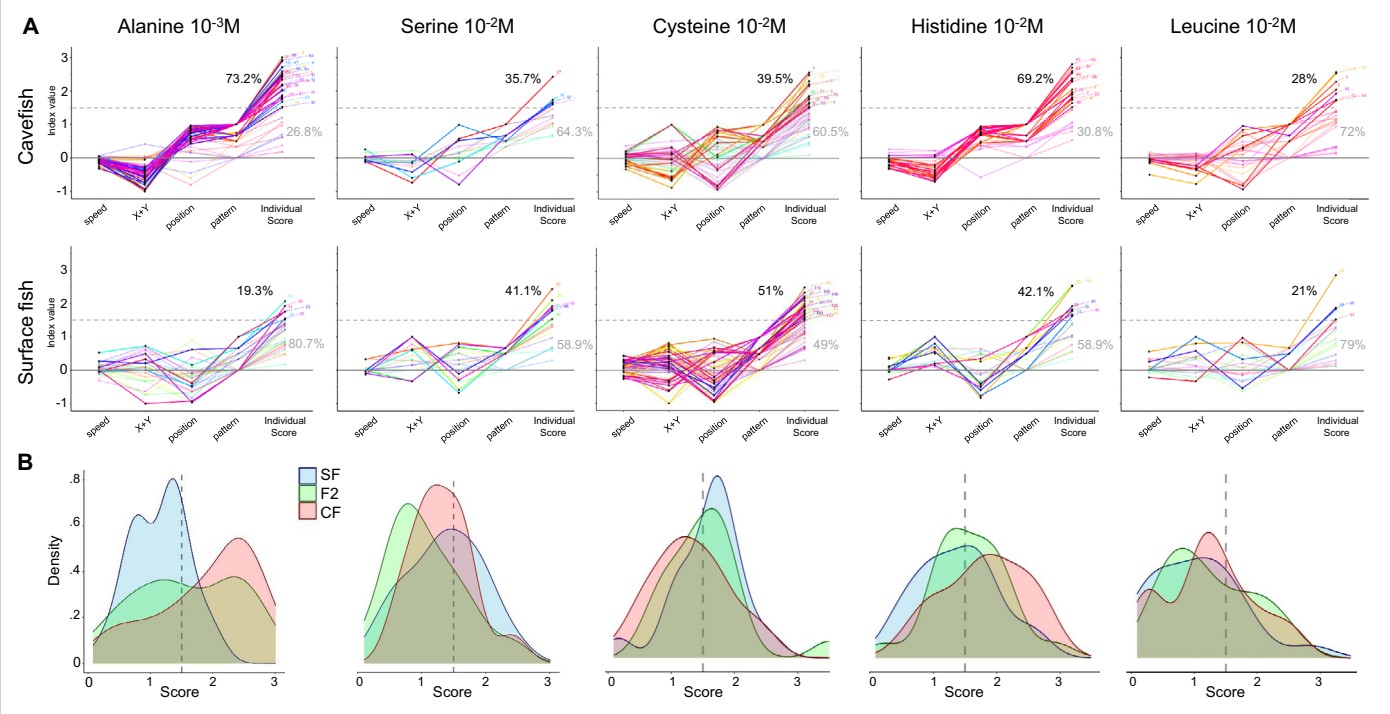

**Figure 7.** Individual olfactory scores of surface fish (SF), cavefish (CF), and F2 hybrids (F2) for different odors. (**A**) Graphs representing index values (i.e. the variation between the 'before' and the 'after' odor condition; the value 0 corresponds to no change) for the four response parameters (speed, thigmotactism in X and Y, position, and swim pattern) used to calculate the total individual olfactory score for each fish (last column of points on each graph). Each fish is depicted by a colored line linking its four different indexes and its final individual olfactory score. The dotted line at value 1.5 indicates the score threshold above which a fish is considered as a responder. The percentages in black and gray indicate the proportion of responders/non-responders, respectively. The colored lines of responders are bright, those of non-responders are pale. Amino acids and concentrations are indicated. Top row: CF; bottom row: SF. (**B**) Distributions of individual olfactory scores of SF (blue), CF (red), and F2 (green) for different odors. The threshold score (1.5) is indicated by a dotted line.

The online version of this article includes the following figure supplement(s) for figure 7:

**Figure supplement 1.** Individual olfactory scores of surface fish (SF), cavefish (CF), and F2 hybrids (F2) in control and experimental conditions.

specific way to high concentrations of lysine, histidine, and leucine. The results of all experiences above are summarised in *Supplementary file 1A*.

For these three odors, CF responses were conspicuous and could include a shift of swim position toward the odor's source, a decrease of back-and-forth swimming activity with decrease in swimming speed, and significant changes in swim patterns at population level (except for leucine) (*Figure 6A–G* and *Figure 6—figure supplement 1A, D, G*; *Figure 6—figure supplement 2* for CA). Regarding SF, changes were restricted to increases in back-and-forth swimming activity for histidine, without change in swimming speed or position in the box (*Figure 6A–G* and *Figure 6—figure supplement 1B, E, H*). These three amino acids did not elicit changes in swimming pattern in SF at population level (*Figure 6E–G*). Finally, F2 also showed specific qualitative (swimming patterns) and quantitative responses to each of these three amino acids (*Figure 6A–G* and *Figure 6—figure supplement 1C, F, I*). Together, these data suggested that CF, SF, and F2 detected and responded in their own and specific way to high concentrations of lysine, histidine, and leucine.

## Behavioral responses to odors at individual level

Analyses as above performed at population level may mask or blur effects or phenotypes in the case when not all individuals respond in a stereotyped manner. As fish did express individual behavioral features in our experimental paradigm, we sought to perform further analyses at individual level, and to calculate individual response scores to the different odors. To do so, and to take into account the different components of the behavioral response, we summed the absolute values of indexes of speed, back-and-forth trips in X and Y, position and pattern changes (see Methods; *Figure 7A*).

The threshold of score for which an individual fish was considered to respond significantly to a given stimulus was set at 1.5, from examination of the individual scores of fish in control conditions after perfusion of water (*Figure 7—figure supplement 1*). Visual inspection of the responses to amino acid odors for fish who had an individual score just above or just below this threshold confirmed that it was accurate.

For alanine $10^{-3}$ M, the representation of individual behavioral responses shows that CF responded in a very stereotyped manner (all lines following the same 'curve') whereas SF responses were more diverse (lines cross) (*Figure 7A*, first column). Moreover, 73% of CF but only 19% of SF had an individual score above threshold – a marked difference that was visible on the distribution of scores (*Figure 7A and B*). F2 fish on the other end had a bimodal distribution of their olfactory scores for alanine $10^{-3}$ M (*Figure 7B*).

For other odors, the difference in olfactory scores of individual CF and SF was less obvious (*Figure 7B* and *Figure 7—figure supplement 1*), suggesting that taking into consideration individual's variations of swimming behavior can unmask responses that are not visible when averaged at population level. For example, in response to cysteine $10^{-2}$ M, SF (51% with score >1.5) and F2 (55% with score >1.5) had similar or slightly better scores than CF (39% with score >1.5) (*Figure 7A and B*, third column). Of note, these results had not shown up in *Supplementary file 1A*, summarizing responses averaged at population level. Noteworthy, SF had mixed, diverse responses to all odors studied (lines crossing on all graphs). Conversely, CF showed non-homogenous responses to serine, histidine, leucine, and lysine but highly stereotyped responses to both alanine and histidine (*Figure 7A* and *Figure 7—figure supplement 1*). A summary of individual response scores to the different odors is given in *Supplementary file 1B*.

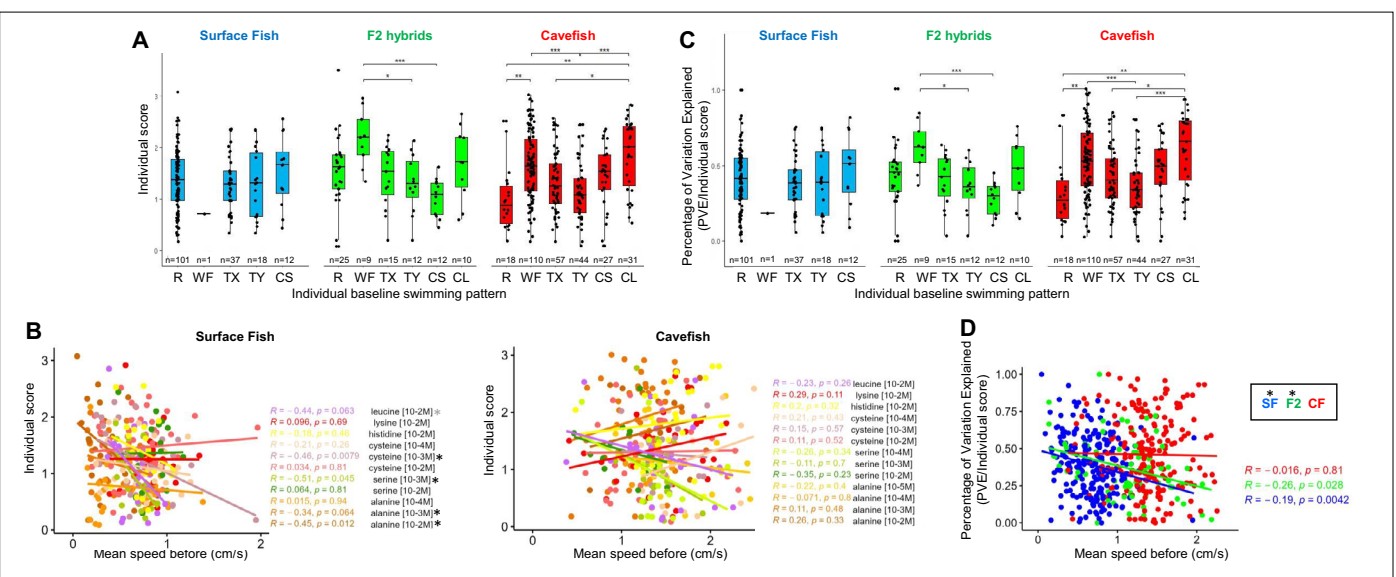

**Figure 8.** The individual olfactory score is related to individual fish swimming personality. (**A**) Box plots showing individual olfactory scores in surface fish (SF) (blue), F2 hybrids (F2) (green), and cavefish (CF) (red) as a function of their individual baseline swimming pattern. As stated in Methods, only amino acid conditions for which more than 40% of fish were responders (score >1.5) are pooled and plotted on this graph. Mann-Whitney two-tailed with Bonferroni post hoc tests were performed on each morphotype. (**B**) Regressions to explore the correlation between baseline swimming speed displayed by individual fish before odor injection and their individual olfactory score. Fish and amino acid type and concentrations are indicated. Linear correlations were calculated with Spearman's rank correlation test followed by Student's t test for the p-values. Conditions for which significant correlation was found are labeled with an asterisk. (**C**) The PVE (percentage of variation explained) on the individual score is plotted as a function of individual baseline behavior, to probe the predictability of the behavioral output as a function of 'swimming pattern personality'. (**D**) The PVE on the individual score is plotted as a function of individual swimming speed, to probe the predictability of the behavioral output as a function of 'locomotor activity personality'. Linear correlations were calculated with Spearman's rank correlation test followed by Student's t test for the p-values. Conditions for which significant correlation was found are labeled with an asterisk. SF (blue), F2 (green), and CF (red).

The online version of this article includes the following figure supplement(s) for figure 8:

**Figure supplement 1.** Relationships between the number of round trip in X and the olfactory score.

## Individual personality and individual behavioral response

The results presented above led us to test whether the 'personality' of each fish, represented by their specific baseline swimming behavior, could influence their ability to respond to odors. To this end, we plotted individual olfactory scores as a function of individual baseline swimming patterns (R, WF, TX, TY, or C). We also examined possible correlations between basal swimming speed or round trip activity and individual olfactory score.

In SF, individual olfactory scores were similar whatever the basal swimming pattern of the fish (*Figure 8A*). However, for several odors, those fish exhibiting lower baseline swimming speed and less round trips in X had better olfactory scores (*Figure 8B* and *Figure 8—figure supplement 1A*; negative correlation). By contrast, in CF, fish exhibiting WF or C as baseline pattern had significantly better individual scores and those swimming randomly (R) had poor scores (*Figure 8A*) – but scores were not correlated whatsoever to baseline swimming speed (*Figure 8B*). Moreover, for alanine $10^{-3}$ M and histidine $10^{-2}$ M, the two odors for which more than 70% of CF showed a significant response (*Figure 7A*), a high number of round trips in X in baseline behavior was also associated to good olfactory scores (in agreement with WF being associated to significant individual response) (*Figure 8—figure supplement 1A*; positive correlation). Finally, F2 that exhibited a WF baseline pattern had better scores than others (like CF) (*Figure 8A*), and swimming speed was poorly correlated to their olfactory performances (not shown). Therefore, swimming patterns of CF and swimming speed of SF seem correlated to their behavioral response scores.

To further assess whether fish baseline swimming parameters are predictive of the behavioral output when presented with an odor, we next calculated how much of the odor response score variation can be explained by 'personality'. We found that the PVE (percentage of variation explained) on the individual olfactory score varies significantly with the baseline swimming pattern in CF (but not in SF) (*Figure 8C*) and with the swimming speed in SF (but not in CF) (*Figure 8D*). Altogether, these analyses strongly suggest that (1) the fish swimming personality has an influence on its response to olfactory stimulation and can be predictive of the response, and (2) the personality parameter that is important to predict a good individual response is not the same in the two *Astyanax* morphs: in SF speed matters, while in CF swimming pattern matters.

## Discussion

We have developed a high-throughput, specific olfaction test, together with a pipeline of analysis allowing assessing qualitatively and quantitatively individual's and population's behaviors, in order to describe and compare responses of blind and sighted *Astyanax* to odorant cues. We discovered that CF, SF, and F2 display different odor preferences and sensitivities and show individual, distinctive, diverse, and specific responses to varying concentrations of the six amino acids tested. Using this novel setup where fish were tested in solo in a rectangular box and during 1 hr (as opposed to testing in groups in a U-shaped box and during 8 min in previous studies; *Blin et al., 2020*; *Blin et al., 2018*; *Hinaux et al., 2016a*), we established that cavefish are bona fide 'alanine specialists' and we analyzed in depth their behavioral responses.

### A setup to probe olfaction in fish with markedly different sensory apparatuses and internal states

There is an inherent difficulty to compare sensory-driven behaviors in cave and surface morphs of *Astyanax*: all their sensory systems have evolved in one way or another. Cavefish have no eyes, but they possess enhanced mechanosensory lateral line and chemosensory gustatory and olfactory organs. To decipher behavioral responses driven by a single of these senses, one needs to control carefully the potential influence of the other sensory modalities. Here, we have recorded unimodal behavioral responses driven exclusively by olfaction by performing experiments in the dark to abrogate vision in sighted fish, and by designing a setup where the delivery of the olfactory stimulus is vibration-free. Moreover, we know from previous studies that gustatory taste buds do not participate in responses to amino acids at the concentrations used because the lesion of the olfactory epithelium abolishes the attraction to high concentrations of alanine, in both SF and CF (*Hinaux et al., 2016a*).

Due to a mutation in their monoamine oxidase enzyme that interferes with the metabolism of brain monoamines (*Elipot et al., 2014*), Pachón cavefish have lower basal cortisol levels, hence lower

basal anxiety than surface fish, when they are long habituated in their home tank (*Pierre et al., 2020*). However, the mutation confers cavefish with a much higher stressability after environmental change, such as the transfer in a novel tank (*Pierre et al., 2020*). Consequently, in order to record relevant behavioral responses to olfactory stimuli in unstressed fish, we used long acclimation times (72 hr) and we tested long habituation times, either 1 hr or 24 hr – as compared to 10 min as usually done in most fish studies including zebrafish or *Astyanax*. We concluded that 1 hr is too short for proper habituation, as the swimming speed and patterns were affected for both SF, CF, and F2 as compared to 24 hr. Importantly, such long, 24 hr habituation periods are susceptible to reveal unperturbed and even novel behaviors: recently applied in a study of social behaviors in cavefish, 3 days of habituation allowed the analysis of behaviors in a familiar environment and could unmask social interactions in the so-called 'asocial' cavefish (*Iwashita and Yoshizawa, 2021*).

## Neurophysiological and molecular considerations

In all vertebrates including fish, odorant molecules are recognized by olfactory receptors expressed at the surface of olfactory sensory neurons, which project onto the olfactory glomeruli in the olfactory bulb with the one receptor: one glomerulus rule (*Axel, 1995*; *Braubach et al., 2012*; *Buck, 2000*; *Kermen et al., 2013*; *Koide et al., 2009*; *Li et al., 2005*; *Yoshihara, 2008*). Moreover, parallel neural pathways in the olfactory circuitry process different types of odorants. In zebrafish, the perception of amino acids is mediated via OlfC/V2R receptors on microvillous sensory neurons that innervate lateral and ventro-medial glomeruli in the bulb. From the periphery to the brain, *Astyanax* surface and cave morphs display some variations in this amino acid signal processing circuitry. SF have 43 and CF have 41 V2R/OlfC receptor genes in their genomes (*Policarpo et al., 2022*), a minor difference that is unlikely to underlie their differences in olfactory capacities and preferences. CF larvae have higher proportions of microvillous neurons than SF, which together with the larger size of their olfactory epithelium and olfactory bulbs may influence their olfactory sensibility (*Blin et al., 2018*). *Astyanax* glomerular organization is unknown.

Interestingly, here we have found that CF are strongly attracted and respond to alanine and histidine, two amino acids which, albeit probably not recognized by the same receptor(s), are processed in the same or very close glomeruli in zebrafish larval olfactory bulbs (*Li et al., 2005*). SF on the other hand seem to show a preference for cysteine (sulfur), the amino acid that is the most potent to evoke electrical responses in the olfactory bulbs of sea breams (*Hubbard et al., 2011*) or hammerhead sharks (*Tricas et al., 2009*), and that elicits a strong aversive behavioral response in larval zebrafish (*Vitebsky et al., 2005*). Therefore, odor preferences have evolved between cavefish and surface fish, as well as between zebrafish and *Astyanax*. In the same line, previously we had reported that chondroitin is a strong attractant for *Astyanax* (*Blin et al., 2020*), whereas it induces freezing and fear behavior in zebrafish (*Mathuru et al., 2012*). Such significant variations in odor preferences or value may be adaptive and relate to the differences in the environmental and ecological conditions in which these different animals live. Of note, we have not found an odor that would be repulsive for *Astyanax* so far, and this may relate to their opportunist, omnivorous, and detritivore regime (*Espinasa et al., 2017*; *Marandel et al., 2020*). However, the reason why Pachón cavefish have become 'alanine specialists' remains a mystery and prompts analysis of the chemical ecology of their natural habitat. Alternatively, specialization for alanine may not need to be specific for an olfactory cue present only, or frequently, or in high amounts in caves. Bat guano for example, which is probably the main source of food in the Pachón cave, must contain and release many amino acids. Enhanced recognition of only one of them – in the present case alanine but evolution may have randomly acted for enhanced recognition of another amino acid – should suffice to confer cavefish with augmented sensitivity to their main source of nutriment.

Cavefish have also evolved regarding olfactory sensitivity. They are able to detect very low concentrations of alanine (aliphatic), hence they have a lower detection threshold than SF. Injection of low concentration alanine $10^{-4}$ M/$10^{-5}$ M (thus $10^{-6}$ M/$10^{-7}$ M in the odorant third of the arena) elicits strong behavioral responses in CF, whereas even higher concentrations of $10^{-3}$ M and $10^{-2}$ M evoke modest responses in SF (zero response at population level, 19% responders at individual level with alanine $10^{-3}$ M). We have performed dose-response experiments for three out of the six amino acids tested and for both morphs. With these, CF also appear capable of detecting low concentrations of cysteine (sulfur). Moreover, they seem to detect better histidine $10^{-2}$ M (polar) as well (69.2% of CF responders,

versus 42.1% of SF), and CF but not SF detected lysine (polar) at the relatively high concentration of $10^{-2}$ M. From these observations, we can predict that the difference between SF and CF probably does not lie in the molecular evolution of their olfactory receptor repertoire, because it is unlikely that all the receptors recognizing diverse types of amino acids have evolved all at once. Rather, we can hypothesize that evolution occurred at the level of the general regulation of odorant receptor genes expression, or at the level of olfactory processing and computation in the bulbs (*Friedrich et al., 2009*), or in higher order brain regions.

## A variety of behavioral responses to olfactory stimulation

Our study suggests that behavioral responses to a unimodal olfactory sensory stimulus are complex and have evolved in cavefish (summary on *Supplementary file 1*). When they detect the odor stimulus, cavefish globally decrease their scanning activity, which might help them to compute and swim up toward the higher values of concentration in the odor plume. This is associated to a change in swimming pattern whereby WF and T are eliminated at the expense of R and C, presumably to facilitate searching. This hypothesis is further supported by their systematic change of position in the box, meaning that they locate efficiently the odor source. Strikingly, surface fish display opposite behavioral responses after odor detection: they rather increase locomotor activity, which corresponds to intense foraging but does not seem optimal to find the odor source, which is confirmed by their lack of change in position of the box (i.e. they may not locate efficiently the odor source). This is also consistent with the fact that at individual level the best SF responders are slow swimmers. Such poor ability to find the odor source may result from testing in the dark. Indeed, SF behaviors are mostly visually driven (but see *Simon et al., 2019*) and they might need multimodal visual+olfactory integration to find food efficiently. These interpretations are consistent with early work showing that, when competing in the dark and with limited amounts of food, SF starve and CF thrive (*Hüppop, 1987*). In sum, CF foraging strategy has evolved in response to the serious challenge of finding food in the dark. Future experiments including functional imaging of brain activity in live animals may reveal the changes in olfactory-driven motor circuits that allowed the evolution of behavioral outputs in cavefish.

Finally, at population level, F2 fish shared some behavioral response traits with both parental morphs, and they were often closer to CF. At individual level their responses were also variable and non-stereotyped (including for alanine $10^{-3}$ M and histidine $10^{-2}$ M, the conditions for which CF showed highly stereotyped responses). Olfactory scores behave as a quantitative trait. The tools we have developed here will allow the future determination of the genetic underpinnings of the evolution of olfactory-driven cavefish behaviors and capabilities.

## A 'personality' for each fish?

Our recordings of several hundred (n=489 total) well-habituated, individual *Astyanax* larvae highlight an often overlooked aspect of fish behavioral analyses: fish may have a 'personality' or 'temperament', which we characterized at the level of their baseline swimming patterns. This hypothesis is strongly supported by the consistency over time, one day after the other, of the favorite swimming pattern or combination of patterns and of the swimming kinetics expressed by individuals when freely swimming without stimulus (most fish were recorded on 4 days distributed over 2 weeks). To our knowledge, this is the first time individual temperament is taken into consideration in *Astyanax* behavioral studies.

In the context of ecology and evolution, the personality in non-human animals was proposed to be organized along five primary axes: sociability, boldness, aggressiveness, exploration, and activity (*Réale et al., 2007*). In fish, studies applying an animal personality approach have focused to resolve variations in physiological and molecular parameters, suggesting a link between phenotype and genotype, or between behavior and transcriptome regulation (*Rey et al., 2021*). Here, we propose that the different temperaments expressed by individual larvae/juveniles could correspond to the genetic diversity present in the natural populations of *Astyanax*, a diversity that we have maintained intentionally along generations in captivity in our fish facility.

Further, we have assessed whether swimming temperaments could influence the way and the extent to which fish respond to an olfactory sensory stimulus. Strikingly, we discovered that baseline swimming patterns and swimming speed do influence fish olfactory responses. In addition, important personality traits that confer positive behavioral responses are not the same in SF and CF. For cavefish, WF and round trips in X (but not speed) are the key parameters. Wall following behavior has

been reported in *Astyanax* and other blind cave-fish species (*Chen et al., 2022*; *Patton et al., 2010*; *Sharma et al., 2009*) and was hypothesized to confer foraging advantages, although this had never been directly tested. Here, it seems to be the case for 6-week-old juveniles in a rectangular box – but the link may be more elusive when considering a fish swimming in a natural, complex environment. We propose that WF and round trips in X are together the expression of an intense exploratory behavior, which allows CF to better cover their swimming space and thus have a higher probability to detect odorant cues – even at very low concentrations because their detection capacities for some amino acids are excellent. By contrast, a slow swimming speed (but not swimming pattern) is the most critical personality parameter for SF. As SF olfactory concentration

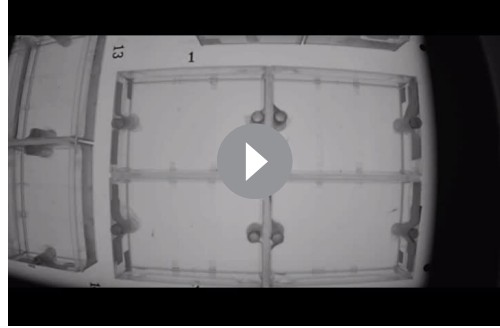

**Video 1.** Infra-red video recorded for 4 CF, showing 30 sec of baseline behavior before odor injection, and 30 sec of olfactory response after injection of alanine 10-3 M on the left.
https://elifesciences.org/articles/92861/figures#video1

detection thresholds are higher than CF, one could imagine that they need more integration time inside the odorant plume to trigger a neuronal and a behavioral response. In any case and importantly, personality parameters that matter in CF and SF to confer good olfactory response scores are distinct. This suggests that the modulation of neuronal circuits underlying both the control of baseline behaviors and olfactory processing has evolved in cavefish.

## Conclusion

Previous studies had shown that, when tested in groups, cavefish larvae as well as adults smell better than their surface fish conspecifics (*Blin et al., 2020*; *Blin et al., 2018*; *Hinaux et al., 2016a*). Here, we have established that it is also true for fish in solo, ruling out the possibility that when in group, one 'good smeller' individual could drive others to respond through (unknown) communication mode. Using individual behavioral tests, we have discovered that both olfactory sensitivity, olfactory preferences, olfactory behavioral responses, and key personality parameters have evolved in cavefish, conferring them with outstanding skills to forage in the darkness of caves.

## Methods
### Fish samples

Laboratory stocks of *A. mexicanus* SF (origin: San Salomon Spring, TX, USA) and CF (Pachón population) were obtained in 2004 from the Jeffery laboratory at the University of Maryland, College Park, MD, USA. Fish were maintained at 22°C with a 12 hr:12 hr light:dark cycle, fed twice daily with dry and live food. SF and CF brood stock were induced to spawn once every 2 weeks by thermal shock at 26°C. An F1 hybrid family was generated by crossing an SF female with a CF Pachón male. F2 were generated by crossing two F1 individuals. Embryos and larvae were reared at 24°C in water produced by our animal house's water production system. All fish used (CF, SF, and F2) were exactly 6 weeks of age on the first day of the experiment. SR's authorization for use of *A. mexicanus* in research is 91-116. Animals were treated according to the French and European regulations for animal testing in research (authorization n°APAFIS#8604).

### Experimental setup

The test consisted of filming individual larvae for 30 min before olfactory stimulation and for a further 30 min afterward (see Supplemental movies *Video 1*). The behavior laboratory is a soundproofed room maintained at a constant temperature of 24°C. To ensure that eyed fish (SF and F2 with eyes) could not use the visual modality during the test (unlike CF and some F2), all experiments (except one) were carried out in the dark. The experimental device was composed of 2 infrared (IR) tables (60×60 cm² each) over which 32 test boxes were placed. A custom-built wooden holder supported

eight IR cameras (DIGITNOW!), located 40 cm above the boxes. Each camera filmed four boxes simultaneously (*Figure 1*).

The individual test boxes (11.5×8.5×4.3 cm³, multiroir #45103BOILAB03) were made of crystal-clear plastic. Odorant solutions (and/or water) were manually injected at the two opposite sides of the box using 1 ml syringes. In order to stop the dispersion wave of injection, and to assure that we always injected at the same right place, we added two injection guides in each box: we glued two plastic tubes (4 cm long and 1 cm in diameter) at the center of the two opposite short sides (Y), 0.2 cm from the bottom. In addition, a net (6.5×6.5 cm²) was attached around each tube and reached the bottom of the box, to prevent fish from passing under the tube during the experiment, thus escaping from the subsequent tracking (*Figure 1*).

### Behavior tests

Larvae were placed individually in a box containing 150 ml of water (clean rearing water), fed with micro-worms (three drops of concentrated *Panagrellus redivivus* solution placed in the center of the box) and subjected to a 12 hr:12 hr day:night cycle for 72 hr acclimation. Then, the water in each box was changed, the boxes were placed on IR tables for 24 hr, and the larvae were fasted (=habituation). The day of testing, for reasons of reproducibility, all experiments started at 11 a.m. The light was switched off, the cameras were switched on, and the experimenter left the room for a 30 min baseline behavior recording period. At t=30 min, the experimenter entered the room and switched on the inactinic red lamps. Solutions were manually injected bilaterally (500 µl of odor/500 µl of water) into the injection guides of each box. The total time needed to perform injections of 32 boxes was 5 min. Then, the experimenter switched off the inactinic lamps and left the room for a 30 min response behavior recording period. At the end of recording, the water was changed and each box was placed back exactly at the same location on IR table for 24 hr and not moved again until the following day of testing (the larvae were not fed). For the series of experiments with a habituation period of 1 hr, the water in the boxes was changed every morning, 1 hr before the start of the test. The typical timeline was as follows: [72 hr acclimation and feeding], week 1 [habituation 24 hr, test day 1, habituation, test day 2, rest and feeding for 3 days], week 2 [habituation, test day 3, habituation, test day 4]. As the larvae were tested on several consecutive days, to ensure that the fish were not learning/associating one side with a scent cue, the odor injection side was reversed each day. The control experiments (unilateral or bilateral injection of 500 µl rearing water or no injection) were always carried out on the day 1 when the larvae were naive. For the 'light vs dark' control experiment, SF were filmed 30 min in normal light and 30 min in the dark.

### Odorant solutions

The odorant solutions were prepared by dilution of amino acids in clean rearing water (L-Alanine #A7627; L-Serine #S4500; L-Lysine #L5501; L-Histidine #H8000; L-Leucine #L8000; and L-Cysteine #168149; all from Sigma-Aldrich). A volume of 500 µl (either water or odor) was injected with a 1 ml syringe (Sigma-Aldrich #Z683531) fitted with a needle (0.8×50 mm², 21 G; B Braun #4665503). Depending on the test, concentration of solutions injected were $10^{-2}$ M or $10^{-3}$ M or $10^{-4}$ M. In a total water volume of 150 ml, with a box virtually separated into three, the volume of water in the odor injection zone is 50 ml. By injecting 500 µl of solution into this zone, we expected to obtain a dilution factor of 100. For example, for a concentration of [$10^{-3}$ M] in the syringe, the concentration in the odor injection zone will therefore be ~[$10^{-5}$ M]. All concentrations given in the article indicate the concentration inside the syringe.

When consecutive injections of an odor or different odors at different concentrations were tested on the same fish, they were performed from the least concentrated to the most concentrated.

### Video editing and tracking

Videos were saved on SD card in AVI format (1208×720 px, 30 fps), then edited using Adobe Premiere Pro V14.0 to be calibrated at 1 hr long without the 5 min injection time. They were then exported into MPG2 format. The tracking software used was TheRealFishTracker V.0.4.0 (https://www.dgp.toronto.edu/~mccrae/projects/FishTracker/), developed and freely available from the University of Toronto. In our hands, it was the only software able to detect properly transparent CF larvae. The parameters used were Confidence Threshold = 10; Mean Filter Size = 1; signed Image = *dark object* and all other

parameters were the default ones. The x/y-scales were drawn and given in cm. The output data file provided 30 X/Y coordinates (in pixels and in cm) per seconds for a 1 hr movie.

## Graphical representations, quantifications, and statistics

Statistical analyses were carried out with R-3.4.2 software (*R Development Core Team, 2016*) using the stats and rstatix libraries, and all graphical representations were designed using the ggplot2, ggrepel, gghighlight, and RGraphics libraries.

## Quantitative parameters

From the output data set, pixel coordinates Xpi and Ypi were first recoded using the box center as 0, Xcmax = 1 on the odor side and Xcmin = –1 on the water side. The same was applied to the Y coordinate with Ycmax = 0.74 and Ycmin = –0.74. Fish position in the box along the X-axis during test was calculated using Xc and Yc at each time point (30/s). Speed between two time points (5 s time step) was calculated, thanks to Pythagoras's theorem using X/Y in cm. Round trip number is the total number of time the fish crosses the 0.5 and –0.5 position (along X-axis) divided by two. Idem for round trips in Y-axis. Quantitative parameter means were all calculated over a 15 min period, from minute 10 to minute 25 for the 'before' period, and from minute 37 to minute 52 for the 'after' period. Means are represented by box plots showing the distribution of individual values of each fish (black dots), the median, the 25th- 75th percentile, and outliers indicated by fish ID number. Paired Mann-Whitney two-tailed tests were performed to compare means of position, speed, or round trips before and after odor injection.

Non-paired Mann-Whitney two-tailed tests were performed to compare means of speed between 1 hr and 24 hr habituation periods represented in *Figure 2C* by box plots.

## Swimming patterns

The swimming activity of the fish in the test box was systematically represented under three different forms, which allow grasping the details and different aspects of the behaviors (see *Figure 2A*). Xc/t coordinates were used with the geom_line() function for the 'Position (X-axis)' graph, (Xc,Yc)/t coordinates were used with the geom_path() function for '2D' top view, and the gg3D package for '2D+time' view.

The determination of baseline swimming patterns and swimming patterns after odor injection was performed manually based on graphical representations such as in *Figure 2A* or *Figure 3A*. Four distinctive baseline behaviors clearly emerged. (1) Random swim (R; defined as haphazard swimming with no clear pattern, covering entirely or partly the surface of the arena). (2) Wall following (WF; defined as the fish continuously following along the four sides of the box and turning around it, in a clockwise or counterclockwise fashion). (3) Large or small circles (C; self-explanatory). (4) Thigmotactism (T, along the X- or the Y-axis of the box; defined as the fish swimming back and forth along one of the four sides of the box). On graphical representations of swimming pattern distributions, we used the following color code: R in blue, WF in red, C in green, T in yellow. Of note, many fish swam according to combination(s) of these four elementary swimming patterns (see descriptions in the legends of the figure supplements showing many examples). To fully represent the diversity and the combinations of swimming patterns used by individual fish, we used an additional color code derived from the 'basic' color code described above and where, e.g., R+WF is purple. The complete combinatorial color code is shown in *Figure 2—figure supplement 2*. Of note, in all figures, the swimming pattern color code does not relate whatsoever with the time color code used in the 2D plus time representation of swimming tracks such as in *Figure 2A*. Swimming patterns (i.e. qualitative variables; compositional dataset) in different conditions were analyzed using Fisher's exact test comparing the quantity and distribution of different swimming patterns between SF/CF/F2 at day 1, or over day 1 to day 4, or between 1 h vs 24 hr habituation period, or before vs after odor injection for each morphotype.

To reinforce our conclusions, swimming patterns in different conditions were also compared using CA, an appropriate method to analyze compositional data (*Greenacre, 2021*). The results are plotted in Figure supplements. The distribution of swimming patterns in SF/CF/F2 in control experiments, or before vs after odor injection for each morphotype confirm the changes (or absence of changes) in behavioral patterns suggested by the colored bar plots in main Figures, with confidence ellipses

overlapping or not overlapping, depending on cases. Of note, CA cannot provide statistical support through calculation of p-values.

### Index and score

For quantitative swimming parameters: indexes of position, speed, and number of round trips changes were calculated using the equation (mean *after* – mean *before*)/(mean *after* + mean *before*) commonly used in olfactory tests in fish (*Choi et al., 2021*; *Koide et al., 2009*; *Wakisaka et al., 2017*). The index can vary from 0 to 1. An index close to 0 indicates that the values of the considered parameter before or after odor injection are close. The higher the absolute value of the index, the greater the difference. When the parameter increases after injection, the index is positive; when the parameter decreases after injection, the index is negative. The *absolute* value of each index will be used for score calculation below.

For qualitative swimming parameter: the index of pattern change was calculated as the number of new patterns observed after stimulation, divided by the total number of patterns observed after odor injection. An index equal to 0 indicates no pattern change after stimulation. The higher the index is, the greater the difference of pattern is.

For each fish, the individual olfactory score for a given odor and concentration was the sum of the *absolute* values of the four indexes: individual score = |position index| + |speed index| + |round trips in X+Y index| + |pattern index|. On *Figure 7A* and *Figure 7—figure supplement 1*, the four parameter's indexes and the total individual fish olfactory scores are represented by dots connected by a colored line for each individual. Those with a score >1.5 are highlighted and considered to display a significant behavioral response to the stimulus. The 1.5 threshold value was arbitrarily determined after the examination of individual olfactory scores in control experiments (*Figure 7—figure supplement 1*) and the observation that the vast majority of fish have individual scores below this value in the absence of specific olfactory stimulus.

On *Figure 8A and C*, to assess the relationship between individual olfactory score and baseline swimming pattern, we pooled results of odors for which more than 40% of individuals have an olfactory score >1.5. So for SF, alanine [$10^{-2}$ M]/cysteine [$10^{-2}$ M]/histidine [$10^{-2}$ M]/serine [$10^{-2}$ M] have been used; for CF, alanine [$10^{-2}$ M]/alanine [$10^{-3}$ M]/alanine [$10^{-4}$ M]/cysteine [$10^{-2}$ M]/cysteine [$10^{-3}$ M]/histidine [$10^{-2}$ M]/lysine [$10^{-2}$ M]; and for F2, alanine [$10^{-3}$ M]/cysteine [$10^{-2}$ M]/histidine [$10^{-2}$ M] (see *Supplementary file 1B*). Mann-Whitney two-tailed with Bonferroni post hoc tests were performed on each morphotype.

Slopes in linear regressions show relationships between mean speed or number of round trips before injection and olfactory score. Linear correlations were calculated with Spearman's rank correlation test followed by Student's t test for the p-values.

## Acknowledgements

We thank Camille Lejeune for her participation in the generation of the SF × CF F1 hybrid line, Maxime Policarpo for preliminary testing of tracking softwares and Rose Tatarsky for interesting discussions. Zootechnical care of our *Astyanax* fish facility was taken in charge by the TEFOR Unit. We are indebted to Joël Attia for invaluable help with statistical analyses during the revision of our manuscript. This work was supported by Equipe FRM (Fondation pour la Recherche Médicale) grant EQU202003010144 to SR.

## Additional information

### Funding

| Funder | Grant reference number | Author |
|---|---|---|
| Fondation pour la Recherche Médicale | EQU202003010144 | Sylvie Rétaux |

The funders had no role in study design, data collection and interpretation, or the decision to submit the work for publication.

## Author contributions
Maryline Blin, Conceptualization, Data curation, Software, Formal analysis, Investigation, Visualization, Methodology; Louis Valay, Software, Investigation, Methodology; Manon Kuratko, Investigation, Methodology; Marie Pavie, Investigation; Sylvie Rétaux, Conceptualization, Resources, Formal analysis, Supervision, Funding acquisition, Validation, Visualization, Methodology, Writing – original draft, Project administration, Writing – review and editing

## Author ORCIDs
Maryline Blin (iD) http://orcid.org/0000-0001-9772-9460
Sylvie Rétaux (iD) https://orcid.org/0000-0003-0981-1478

## Ethics
Animals were treated according to the French and European regulations for animal testing in research (Ethics Committee protocol authorization n°APAFIS#8604). SR's authorization for use of Astyanax mexicanus in research is 91-116.

Joint public review: https://doi.org/10.7554/eLife.92861.3.sa1
Author response https://doi.org/10.7554/eLife.92861.3.sa2

---

# Additional files

### Supplementary files
• MDAR checklist

• Supplementary file 1. Summary of the results. (A) Population level summary. For each amino acid and each concentration tested, arrows indicate whether the considered parameter has changed (increased or decreased). ns indicates no significant change, - indicates the condition was not tested in the fish type (SF in blue, CF in red, F2 in green). (B) Individual level summary. The percentage of fish displaying an individual olfactory score superior to 1.5 is indicated.

### Data availability
All data generated or analysed during this study are included in the manuscript and supporting files; source data files have been provided. The R script used to recode raw data into corrected coordinates, to calculate means and to draw 2D and 2D+time graphs is available on GitHub (https://github.com/Maryline-Blin/Astyanax-swimming-pattern; copy archived at *Maryline-Blin, 2024*). *Figure 2—source data 1*, *Figure 2—source data 2*, *Figure 4—source data 1*, *Figure 5—source data 1*, and *Figure 6—source data 1* correspond to raw data (.txt) presented and analyzed in this paper. For each type, the files provide the ID numbers of the fish, condition (day 1 to day 4 or odor or control type), means of position, speed, X and Y round trips and behavior for periods 10–25 min and 37–52 min, as well as the calculated individual olfactory score.

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
