## [Editor Report · eLife assessment]

In this **important** paper, Blin and colleagues develop a high-throughput behavioral assay to test spontaneous swimming and olfactory preference in individual Mexican cavefish larvae. The authors present **compelling** evidence that the surface and cave morphs of the fish show different olfactory preferences and odor sensitivities and that individual fish show substantial variability in their spontaneous activity that is relevant for olfactory behaviour. The paper will be of interest to neurobiologists working on the evolution of behaviour, olfaction, and the individuality of behaviour.

---

## [Referee Report · Joint public review]

Summary:

The paper explores chemosensory behaviour in surface and cave morphs and F2 hybrids in the Mexican cave fish Astyanax mexicanus. The authors develop a new behavioural assay for the long-term imaging of individual fish in a parallel high-throughput setup. The authors first demonstrate that the different morphs show different basal exploratory swimming patterns and that these patterns are stable for individual fish. Next, the authors test the attraction of fish to various concentrations of alanine and other amino acids. They find that the cave morph is a lot more sensitive to chemicals and shows directional chemotaxis along a diffusion gradient of amino acids. Surface fish, although can detect the chemicals, do not show marked chemotaxis behaviour and have an overall lower sensitivity. These differences have been reported previously but the authors report longer-term observations on many individual fish of both morphs and their F2 hybrids. The data also indicate that the observed behaviour is a quantitative genetic trait. The approach presented will allow the mapping of genes contribution to these traits. The work will be of general interest to behavioural neuroscientists and those interested in olfactory behaviours and the individual variability in behavioural patterns.

Strengths:

The authors provide a large dataset of swimming behaviour for surface fish and cave fish and also their F2 hybrids, demonstrating large differences in chemosensory behaviour and indicating that this is a quantitative genetic trait.

One strength of the paper is the development of a new and improved setup for the behavioural imaging of individual fish for extended periods and under chemosensory stimulation. The authors show that cave fish need up to 24 h of habituation to display a behavioural pattern that is consistent and unlikely to be due to the stressed state of the animals. The setup also uses relatively large tanks that allows the build-up of chemical gradients.

With their new system, the authors could generate cleaner results without mechanical disturbances. The authors characterize multiple measurements to score the odour response behaviours and also developed a new personality analysis. Their conclusion that cave fish evolved as a specialist to sense alanine and histidine among 6 tested amino acids was well supported by their data.

Weaknesses:

Further work will be needed to pinpoint the nature of the genetic changes and neurobiological mechanisms that underlie the differences between the two forms and the large individual variation of behaviours.

The authors did not measure the concentrations of alanine and other amino acids in the local cave water and surface water.

---

## [Author Response]

The following is the authors’ response to the original reviews.

**eLife assessment**
In this important paper, Blin and colleagues develop a high-throughput behavioral assay to test spontaneous swimming and olfactory preference in individual Mexican cavefish larvae. The authors present compelling evidence that the surface and cave morphs of the fish show different olfactory preferences and odor sensitivities and that individual fish show substantial variability in their spontaneous activity that is relevant for olfactory behaviour. The paper will be of interest to neurobiologists working on the evolution of behaviour, olfaction, and the individuality of behaviour.
**Public Reviews:**

**Reviewer #1 (Public Review):**
Summary:The authors posed a research question about how an animal integrates sensory information to optimize its behavioral outputs and how this process evolved. Their data (behavioral output analysis with detailed categories in response to the different odors in different concentrations by comparing surface and cave populations and their hybrid) partially answer this tough question. They built a new low-disturbance system to answer the question. They also found that the personality of individual fish is a good predictor of behavioral outputs against odor response. They concluded that cavefish evolved to specialize their response to alanine and histidine while surface fish are more general responders, which was supported by their data.Strengths:With their new system, the authors could generate clearer results without mechanical disturbances. The authors characterize multiple measurements to score the odor response behaviors, and also brought a new personality analysis. Their conclusion that cavefish evolved as a specialist to sense alanine and histidine among 6 tested amino acids was well supported by their data.Weaknesses:The authors posed a big research question: How do animals evolve the processes of sensory integration to optimize their behavioral outputs? I personally feel that, to answer the questions about how sensory integration generates proper (evolved) behavior, the authors at least need to show the ecological relevance of their response. For the alanine/histidine preference in cavefish, they need data for the alanine and other amino acid concentrations in the local cave water and compare them with those of surface water.

We agree with the reviewer. This is why, in the Discussion section, we had written: “…Such significant variations in odor preferences or value may be adaptive and relate to the differences in the environmental and ecological conditions in which these different animals live. However, the reason why Pachón cavefish have become “alanine specialists” remains a mystery and prompts analysis of the chemical ecology of their natural habitat. Of note, we have not found an odor that would be repulsive for Astyanax so far, and this may relate to their opportunist, omnivorous and detritivore regime (Espinasa et al., 2017; Marandel et al., 2020).” This is also why we currently develop field work projects aimed at clarifying this question. However, such experiments and analyses are challenging, practically and technically. We hope we can reach some conclusions in the future.

To complete the discussion we have also added an important hypothesis: “Alternatively, specialization for alanine may not need to be specific for an olfactory cue present only, or frequently, or in high amounts in caves. Bat guano for example, which is probably the main source of food in the Pachón cave, must contain many amino acids. Enhanced recognition of one of them - in the present case alanine but evolution may have randomly acted for enhanced recognition of another amino acid – should suffice to confer cavefish with augmented sensitivity to their main source of nutriment.”

Also, as for "personality matters", I read that personality explains a large variation in surface fish. Also, thigmotaxis or wall-following cavefish individuals are exceeded to respond well to odorants compared with circling and random swimming cavefish individuals. However, I failed to understand the authors' point about how much percentages of the odorant-response variations are explained (PVE) by personality. Association ( = correlation) was good to show as the authors presented, but showing proper PVE or the effect size of personality to predict the behavioral outputs is important to conclude "personality is matter"; otherwise, the conclusion is not so supported.From the above, I recommend the authors reconsider the title also their research questions well. At this moment, I feel that the authors' conclusions and their research questions are a little too exaggerated, with less supportive evidence.

Thank you for this interesting suggestion, which we have fully taken into consideration. We have therefore now calculated and plotted PVE (the percentage of variation explained on the olfactory score) as a function of swimming speed or as a function of swimming pattern. The results are shown in modified Figure 8 of our revised ms and they suggest that the personality (here, swimming patterns or swimming speed) indeed predicts the olfactory response skills. Therefore, we would like to keep our title as we provide support for the fact that “personality matters”.

Also, for the statistical method, Fisher's exact test is not appropriate for the compositional data (such as Figure 2B). The authors may quickly check it at https://en.wikipedia.org/wiki/Compositional_data or https://www.annualreviews.org/doi/pdf/10.1146/annurev-statistics-042720-124436.The authors may want to use centered log transformation or other appropriate transformations (Rpackage could be: https://doi.org/10.1016/j.cageo.2006.11.017). According to changing the statistical tests, the authors' conclusion may not be supported.

Actually, in most cases, the distributions are so different (as seen by the completely different colors in the distribution graphs) that there is little doubt that swimming behaviors are indeed different between surface and cavefish, or between ‘before’ and ‘after’ odor stimulation. However, it is true that Fisher’s exact test is not fully appropriate because data can be considered as compositional type. For this kind of data, centered log transformation have been suggested. However, our dataset contains many zeros, and this is a case where log transformations have difficulty handling.

To help us dealing with our data, the reviewer proposed to consider the paper by Greenacre (2021)(https://www.annualreviews.org/doi/pdf/10.1146/annurev-statistics-042720-124436). In his paper,Greenacre clearly wrote: "Zeros in compositional data are the Achilles heel of the logratio approach (LRA)."

Therefore, we have now tested our data using CA (Correspondence Analysis), that can deal with table containing many zeros and is a trustable alternative to LRA (Cook-Thibeau, 2021; Greenacre, 2011).

The results of CA analysis are shown in Supplemental figure 8 and they fully confirm the difference in baseline swimming patterns between morphs as well as changes (or absence of changes) in behavioral patterns after odor stimulation suggested by the colored bar plots in main figures, with confidence ellipses overlapping or not overlapping, depending on cases. Therefore, the CA method fully confirms and even strengthens our initial interpretations.

Finally, we have kept our initial graphical representation in the ms (color-coded bar plots; the complete color code is now given in Suppl. Fig7), and CA results are shown in Suppl. Figure 8 and added in text.

**Reviewer #2 (Public Review):**
In their submitted manuscript, Blin et al. describe differences in the olfactory-driven behaviors of river-dwelling surface forms and cave-dwelling blind forms of the Mexican tetra, Astyanax mexicanus. They provide a dataset of unprecedented detail, that compares not only the behaviors of the two morphs but also that of a significant number of F2 hybrids, therefore also demonstrating that many of the differences observed between the two populations have a clear (and probably relatively simple) genetic underpinning.To complete the monumental task of behaviorally testing 425 six-week-old Astyanax larvae, the authors created a setup that allows for the simultaneous behavioral monitoring of multiple larvae and the infusion of different odorants without introducing physical perturbations into the system, thus biasing the responses of cavefish that are particularly fine-tuned for this sensory modality. During the optimization of their protocol, the authors also found that for cave-dwelling forms one hour of habituation was insufficient and a full 24 hours were necessary to allow them to revert to their natural behavior. It is also noteworthy that this extremely large dataset can help us see that population averages of different morphs can mask quite significant variations in individual behaviors.Testing with different amino-acids (applied as relevant food-related odorant cues) shows that cavefish are alanine- and histidine-specialists, while surface fish elicit the strongest behavioral responses to cysteine. It is interesting that the two forms also react differently after odor detection: while cave-dwelling fish decrease their locomotory activity, surface fish increase it. These differences are probably related to different foraging strategies used by the two populations, although, as the observations were made in the dark, it would be also interesting to see if surface fish elicit the same changes in light as well.

Thank you for these nice comments.

Further work will be needed to pinpoint the exact nature of the genetic changes that underlie the differences between the two forms. Such experimental work will also reveal how natural selection acted on existing behavioral variations already present in the SF population.

Yes. Searching for genetic underpinnings of the sensory-driven behavioral differences is our current endeavor through a QTL study and we should be able to report it in the near future.

It will be equally interesting, however, to understand what lies behind the large individual variation of behaviors observed both in the case surface and cave populations. Are these differences purely genetic, or perhaps environmental cues also contribute to their development? Does stochasticity provided by the developmental process has also a role in this? Answering these questions will reveal if the evolvability of Astyanax behavior was an important factor in the repeated successful colonization of underground caves.

Yes. We will also access (at least partially) responses to most of these questions in our current QTL study.

**Reviewer #3 (Public Review):**
Summary:The paper explores chemosensory behaviour in surface and cave morphs and F2 hybrids in the Mexican cavefish Astyanax mexicanus. The authors develop a new behavioural assay for the longterm imaging of individual fish in a parallel high-throughput setup. The authors first demonstrate that the different morphs show different basal exploratory swimming patterns and that these patterns are stable for individual fish. Next, the authors test the attraction of fish to various concentrations of alanine and other amino acids. They find that the cave morph is a lot more sensitive to chemicals and shows directional chemotaxis along a diffusion gradient of amino acids. For surface fish, although they can detect the chemicals, they do not show marked chemotaxis behaviour and have an overall lower sensitivity. These differences have been reported previously but the authors report longer-term observations on many individual fish of both morphs and their F2 hybrids. The data also indicate that the observed behavior is a quantitative genetic trait. The approach presented will allow the mapping of genes' contribution to these traits. The work will be of general interest to behavioural neuroscientists and those interested in olfactory behaviours and the individual variability in behavioural patterns.Strengths:A particular strength of this paper is the development of a new and improved setup for the behavioural imaging of individual fish for extended periods and under chemosensory stimulation. The authors show that cavefish need up to 24 h of habituation to display a behavioural pattern that is consistent and unlikely to be due to the stressed state of the animals. The setup also uses relatively large tanks that allow the build-up of chemical gradients that are apparently present for at least 30 min.The paper is well written, and the presentation of the data and the analyses are clear and to a high standard.

Thank you for these nice comments.

Weaknesses:One point that would benefit from some clarification or additional experiments is the diffusion of chemicals within the behavioural chamber. The behavioural data suggest that the chemical gradient is stable for up to 30 min, which is quite surprising. It would be great if the authors could quantify e.g. by the use of a dye the diffusion and stability of chemical gradients.

OK. We had tested the diffusion of dyes in our previous setup and we also did in the present one (not shown). We think that, due to differences of molecular weight and hydrophobicity between the tested dyes and the amino acid molecules we are using, their diffusion does not constitute a proper read-out of actual amino acid diffusion. We anticipate that amino acid diffusion is extremely complex in the test box, possibly with odor plumes diffusing and evolving in non-gradient patterns, in the 3 dimensions of the box, and potentially further modified by the fish swimming through it, the flow coming from the opposite water injection side and the borders of the box. This is the reason why we have designed the assay with contrasting “odor side” and “water control side”. Moreover, our question here is not to determine the exact concentration of amino acid to which the fish respond, but to compare the responses in cavefish, surface fish and F2 hybrids. Finally and importantly, we have performed dose/response experiments whereby varying concentrations have been presented for 3 of the 6 amino acids tested, and these experiments clearly show a difference in the threshold of response of the different morphs.

The paper starts with a statement that reflects a simplified input-output (sensory-motor) view of the organisation of nervous systems. "Their brains perceive the external world via their sensory systems, compute information and generate appropriate behavioral outputs." The authors' data also clearly show that this is a biased perspective. There is a lot of spontaneous organised activity even in fish that are not exposed to sensory stimulation. This sentence should be reworded, e.g. "The nervous system generates autonomous activity that is modified by sensory systems to adapt the behavioural pattern to the external world." or something along these lines.

Done

**Recommendations for the authors:**

**Reviewer #1 (Recommendations For The Authors):**
In addition to my comments in the "weakness" section above, here are my other comments.How many times fish were repeatedly assayed and what the order (alanine followed by cysteine, etc) was, is not clear (Pg 24, Materials and Methods). I am afraid that fish memorize the prior experience to get better/worse their response to the higher conc of alanine, etc. Please clarify this point.

Many fish were tested in different conditions on consecutive days, indeed. Most often, control experiments (eg, water/nothing; water/water; nothing/nothing) were followed by odor testing. In such cases, there is no risk that fish memorize prior experience and that such previous experience interferes with response to odor. In other instances, fish were tested with a low concentration of one amino acid, followed by a high concentration of another amino acid, which is also on the safe side. Of note, on consecutive days, the odors were always perfused on alternate sides of the test box, to avoid possibility of spatial memory. Finally, in the few cases where increasing concentrations of the same amino acids were perfused consecutively, (1) they were perfused on alternate sides, (2) if the fish does not detect a low concentration below threshold / does not respond, then prior experience should not interfere for responding to higher concentrations, and (3) we have evidence (unpublished, current studies) that when a fish is given increasing concentrations of the same amino acid above detection threshold, then the behavioral response is stable and reproducible (eg does not decrease or increase).

Minor points:Thygmotaxis and wall following.Classically, thigmotaxis and wall following are treated as the same (sharma et al., 2009; https://pubmed.ncbi.nlm.nih.gov/19093125/) but the authors discriminate it in thigmotaxis at X-axis and Y-axis because fish repeatedly swam back and forth on x-axis wall or y-axis wall. I understand the authors' point to discriminate WF and T but present them with more explanations (what the differences between them) in the introduction and result sections.

Done

Pg5 "genetic architecture" in the introduction."Genetic architecture" analysis needs a more genomic survey, such as GWAS, QTL mapping, and Hi-C. Phenotype differences in F2 generation can be stated as "genetic factor(s)" "genetic component(s)", etc. please revise.

Done

Pg10 At the serine treatment, the authors concluded that "...suggesting that their detection threshold for serine is lower than for alanine." I believe that the 'threshold for serine is higher' according to the authors' data. Their threshold-related statement is correct in Pg21 "as SF olfactory concentration detection threshold are higher than CF,..." So the statement on page 10 is a just mistake, I think. Please revise.

Done (mistake indeed)

Pg11 After explaining Fig5, the statement "In sum, the responses of the different fish types to different concentrations of different amino acids were diverse and may reflect complex, case-bycase, behavioral outputs" does not convey any information. Please revise.

OK. Done : “In sum, the different fish types show diverse responses to different concentrations of different amino acids.”

For the personality analysis (Fig 7)The index value needs more explanation. I read the materials and methods three times but am still confused. From the equation, the index does not seem to exceed 1.0, unless the "before score" was a negative value, and the "after score" value was positive. I could not get why the authors set a score of 1.5 as the threshold for the cumulative score of these different behavior index values ( = individual score). Please provide more description. Currently, I am skeptical about this index value in Fig 7.

Done, in results and methods.

Pg15 the discussion sectionPlease discuss well the difference between the authors' finding (cavefish respond 10^-4M for position and surface fish responded 10^-4 for thig-Y; Fig 4AB), and those in Hinaux et al. 2016 (cavefish responded 10^-10M alanine but surface fish responded 10^-5M or higher). It seems that surface fish could respond to the low conc of alanine as cavefish do, which is opposed to the finding in Hinaux 2016.

The increase in NbrtY at population level for surface fish with 10-4M alanine (~10-6M in box) was most probably due to only a few individuals. Contrarily to cavefish, all other parameters were unchanged in surface fish for this concentration. Moreover, at individual level, only 3.2% of surface fish had significant olfactory scores (to be compared to 81.3% for cavefish). Thus, we think that globally this result does not contradict our previous findings in Hinaux et al (2016), and solely represent the natural, unexplained variations inherent to the analysis of complex animal behaviors – even when we attempt to use the highest standards of controlled conditions.

Of note, in the revised version, we have now included a full dose/response analysis for alanine concentration ranging from 10-2M to 10-10M, on cavefish. Alanine 10-5M has significant effects (now shown in Suppl Fig2 and indicated in text; a column has been added for 10-5M in Summary Table 1). Lower concentrations have milder effects (described in text) but confirm the very low detection threshold of cavefish for this amino acid.

Pg19, "In sum, CF foraging strategy has evolved in response to the serious challenge of finding food in the dark"My point is the same as explained in the 'weakness' section above: how this behavior is effective in the cave life, if they conclude so? Please explain or revise this statement.

The present manuscript reports on experiments performed in “artificial” and controlled laboratory conditions. We are fully aware that these conditions are probably distantly related to conditions encountered in the wild. Note that we had written in original version (page 20) “…for 6-week old juveniles in a rectangular box - but the link may be more elusive when considering a fish swimming in a natural, complex environment.” As the reviewer may know, we also perform field studies in a more ethological approach of animal behaviors, thus we may be able to discuss this point more accurately in the future.

Pg20 "To our knowledge, this is the first time individual variations are taken into consideration inAstyanax behavioral studies."This is wrong. Please see Fernandes et al., 2022. (https://pubmed.ncbi.nlm.nih.gov/36575431/).

OK. The sentence is wrong if taken in its absolute sense, i.e., considering inter-individual variations of a given parameter (e.g., number of neuromasts per individual or number of approaches to vibrating rod in Fernandez et al, 2022). In this same sense, Astyanax QTL studies on behaviors in the past also took into account variations among F2 individuals. Here, we wanted to stress that personality was taken into consideration. The sentence has been changed: “To our knowledge, this is the first time individual temperament is taken into consideration in Astyanax behavioral studies.”

Figure 2B and others.The order of categories (R, R-TX, etc) should match in all columns (SF, F2, and CF). Currently, the category orders seem random or the larger ratio categories at the bottom, which is quite difficult to compare between SF, F2, and CF. Also, the writings in Fig 2A (times, Y-axis labels, etc), and the bargraphs' writings are quite difficult to read in Fig 2B, Fig 3B 4H, 5GN, 6EFG. Also, no need to show fish ID in Fig 2C in the current way, but identify the fish data points of the fish in Fig 2D (SF#40, CF#65, and F2#26) in Fig 2C if the authors want to show fish ID numbers in the boxplots. Fish ID numbers in other boxplot figures are recommended to be removed too.

We have thought a lot on how to best represent the distributions of swimming patterns in graphs such as Fig 2B and others. The difficulty is due to the existence of many combinations (33 possibilities in total, see new Suppl Fig7), which are never the same in different plots/conditions because individual tested fish are different. We decided that that the best way was to represent, from bottom to top, the most used to the less used swimming patterns, and to use a color code that matches at best the different combinations. It was impossible to give the full color code on each figure, therefore it was simplified, and we believe that the results are well conveyed on the graphs. We would like to keep it as it is. To respond (partially) to the reviewer’s concern, we have now added a full color code description in a new Supplemental Figure 7 (associated to Methods).

Size of lettering has been modified in all pattern graphs like Fig2A. Thanks for the suggestion, it reads better now.

Finally, we would like to keep the fish ID numbers because this contributes to conveying the message of the paper, that individuality matters.

Raw data files were not easy to read in Excel or LibreOffice. Please convert them into the csv format to support the rigor in the authors' conclusion.

We do not understand this request. Our very large dataset must be analysed with R, not excel for stats or for plotting and pattern analysis. However, raw data files can be opened in excel with format conversion.

**Reviewer #2 (Recommendations For The Authors):**
I think most of the experimental procedures (with few exceptions, see below) are well-defined and nicely described, so the majority of my suggestions will be related to the visualization of the data. I think the authors have done a great job in presenting this complex dataset, but there are still some smaller tweaks that could be used to increase the legibility of the presented data.First and perhaps foremost, a better definition of the swimming pattern subsets is needed. I have no problem understanding the main behavioral types, but whereas the color codes for these suggest that there is continuous variance within each pattern, it is not clear (at least to me), what particular aspect(s) of the behaviors vary. Also, whereas the sidebars/legends suggest a continuum within these behaviors, the bar charts themselves clearly present binned data. I did not find a detailed description of how the binning was done. As this has been - according the Methods section - a manual process, more clarity about the details of the binning would be welcome. I would also suggest using binned color codes for the legends as well.

Done, in Results and Methods. We hope it is now clear that there is no “continuum”, rather multiple combinations of discrete swimming patterns. The gradient aspect in color code in figures has been removed to avoid the idea of continuum. According to the chosen color code, WF is in red, R in blue, T in yellow and C in green. Then, combination are represented by colors in between, for example, R+WF is purple. We have now added a full color code description for the swimming patterns and their combinations in a new Supplemental Figure 7 (associated to Methods).

Also, to better explain the definition of the swimming patterns and the graphical representation, it now reads (in Methods):

“The determination of baseline swimming patterns and swimming patterns after odor injection was performed manually based on graphical representations such as in Figure 2A or Figure 3A. Four distinctive baseline behaviors clearly emerged: random swim (R; defined as haphazard swimming with no clear pattern, covering entirely or partly the surface of the arena), wall following (WF; defined as the fish continuously following along the 4 sides of the box and turning around it, in a clockwise or counterclockwise fashion), large or small circles (C; self explanatory), and thigmotactism (T, along the X- or the Y-axis of the box; defined as the fish swimming back and forth along one of the 4 sides of the box). On graphical representations of swimming pattern distributions, we used the following color code: R in blue, WF in red, C in green, T in yellow. Of note, many fish swam according to combination(s) of these four elementary swimming patterns (see descriptions in the legends of Supplemental figures, showing many examples). To fully represent the diversity and the combinations of swimming patterns used by individual fish, we used an additional color code derived from the “basic” color code described above and where, for example R+WF is purple. The complete combinatorial color code is shown in Suppl. Fig7.”

It would be also easier to comprehend the stacked bar charts, presenting the particular swimming patterns in each population, if the order of different swimming patterns was the same for all the plots (e.g. the frequency of WF always presented at the bottom, R on the top, and C and T in the middle). This would bring consistency and would highlight existing differences between SF, CF, and F2s. Furthermore, such a change would also make it much easier to see (and compare) shifts in behaviors.

We have thought a lot on how to best represent the distributions of swimming patterns in graphs such as Fig 2B and others. The difficulty is due to the existence of many combinations, which are never the same in different plots/conditions because the individual fish tested are different. We decided to keep it as it currently stands, because we think re-doing all the graphs and figures would not significantly improve the representation. In fact, we think that the differences between morphs (dominant blue in SF, dominant red in CF) and between conditions (bar charts next to each other) are easy to interpret at first glance in the vast majority of cases. Moreover, they are now completed by CA analyses (Suppl Figure 8).

While the color coding of the timeline in the "3D" plots presented for individual animals is a nice feature, at the moment it is slightly confusing, as the authors use the same color palette as for the stacked bar charts, representing the proportionality of the particular swimming patterns. As the y-axis is already representing "time" here, the color coding is not even really necessary. If the authors would like to use a color scheme for aesthetic reasons, I would suggest using another palette, such as "grey" or "viridis".

We would like to keep the graphical aspect of our figures as they are, for aesthetic reasons. To avoid confusion with stacked bar chart color code, we have added a sentence in Methods and in the legend of Figure 2, where the colors first appear:

“The complete combinatorial color code is shown in Suppl. Figure 7. Of note, in all figures, the swimming pattern color code does not relate whatsoever with the time color code used in the 2D plus time representation of swimming tracks such as in Figure 2A”.

I would also suggest changing the boxplots to violin-plots. Figure 7 clearly shows bimodality for F2 scores (something, as the authors themselves note, not entirely surprising given the probably poligenic nature of the trait), but looking at SF and CF scores I think there are also clear hints for non-normal distributions. If non-normal distribution of traits is the norm, violin-plots would capture the variance in the data in a more digestible way. (The existence of differently behaving cohorts within the population of both SF and CF forms would also help to highlight the large pre-existing variance, something that was probably exploited by natural selection as well, as mentioned briefly in the Discussion by the authors, too.)

The bimodal distribution of scores shown by F2s in Figure 7B is indeed probably due to the polygenic nature of the trait. However, such distribution is rather the exception than the norm. Moreover, the boxplot representations we have used throughout figures include all the individual points, and outliers can be identified as they have the fish ID number next to them. This allows the reader to grasp the variance of the data. Again, redoing all graphs and figures would constitute a lot of work, for little gain in term of conveying the results. Therefore, we choose not to change the boxplot for violin plots.

The summary data of individual scores in Table 1B shows some intriguing patterns, that warrant a bit further discussion, in my opinion. For example, we can see opposite trends in scores of SF and CF forms with increasing alanine concentration. Is there an easy explanation for this? Also, in the case of serine, the CF scores do not seem to respond in a dose-dependent manner and puzzlingly at 10^(-3)M serine concentration F2 scores are above those of both grandparental populations.

That is true. However, we have no simple explanation for this. To begin responding to this question, we have now performed full dose/responses expts for alanine (concentrations tested from 10-2M to 10-10M on cavefish; confirm that CF are bona fide “alanine specialists”) and for serine (10-2M to 104M tested on both morphs; confirm that both morphs respond well to this amino acid). These complementary results are now included in text and figures (partially) and in the summary table 1.

If anything is known about this, I would also welcome some discussion on how thigmotactic behavior, a marker of stress in SF, could have evolved to become the normal behavior of CF forms, with lower cortisol levels and, therefore lower anxiety.

We actually think thigmotactism is a marker of stress in both morphs. See Pierre et al, JEB 2020, Figure S3A: in both SF and CF thigmotaxis behavior decreases after long habituation times. In our hands, the only difference between the two morphs is that surface fish (at 5 month of age) express stress by thigmotactism but also freezing and rapid erratic movements, while cavefish have a more restricted stress repertoire.

This is why in the present paper we have carefully made the distinction between thigmotactism ( = possible stress readout) and wall following ( = exploratory behavior). Our finding that WF and large circles confers better olfactory response scores to cavefish is in strong support of the different nature of these two swimming patterns. Then, why is swimming along the 4 walls of a tank fundamentally different from swimming along one wall? The question is open, although the number of changes of direction is probably an important parameter: in WF the fish always swims forward in the same direction, while in T the fish constantly changes direction when reaching the corner of the tank – which is similar to erratic swim in stressed surface fish.

Finally two smaller suggestions:When referring to multiple panels on the same figure it would be better to format the reference as "Figure 4D-G" instead of "Figure 4DEFG";

Done

On page 4, where the introduction reads as "although adults have a similar olfactory rosette with 2025 lamellae", in my opinion, it would be better to state that "while adults of the two forms have a similar olfactory rosette with 20-25 lamellae".

Done

**Reviewer #3 (Recommendations For The Authors):**
Consider moving Figure 3 to be a supplement of Figure 4. This figure shows a water control and therefore best supplements the alanine experiment.

We would like to keep this figure as a main figure: we consider it very important to establish the validity of our behavioral setup at the beginning of the ms, and to establish that in all the following figures we are recording bona fide olfactory responses.

"sensory changes in mecano-sensory and gustatory systems " - mechano-sensory.

Done

Figure 2 legend: "(3) the right track is the 3D plus time (color-coded)" - shouldn't it be 2D plus time or 3D (x,y, time).

True! Thanks for noting this, corrected.

Figure 4 legend "E, Change in swimming patterns" should be H.

Done

"suggesting that their detection threshold for serine is lower than for alanine" - higher?

Done

In the behavioural plots, I assume that the "mean position" value represents the mean position along the X-axis of the chamber - this should be clarified and the axis label updated accordingly.

That is correct and has been updated in Methods and Figures and legends.

"speed, back and forth trips in X and Y, position and pattern changes (see Methods; Figure 7A)." - here it would be helpful to add an explanation like "to define an olfactory score for individual fish."

This has been changed in Results and more detailed explanations on score calculations are now given in Methods.

"possess enhanced mecanosensory lateral line" - mechanosensory.

Done